# Uncertainty-based out-of-distribution detection requires suitable function space priors

## Abstract

The need to avoid confident predictions on unfamiliar data has sparked interest in *out-of-distribution* (OOD) detection. It is widely assumed that Bayesian neural networks (BNNs) are well suited for this task, as the endowed epistemic uncertainty should lead to disagreement in predictions on outliers. In this paper, we question this assumption and show that proper Bayesian inference with function space priors induced by neural networks does not necessarily lead to good OOD detection. To circumvent the use of approximate inference, we start by studying the infinite-width case, where Bayesian inference can be exact due to the correspondence with Gaussian processes. Strikingly, the kernels induced under common architectural choices lead to distributions over functions which cause predictive uncertainties that do not reflect the underlying data generating process and are therefore unsuited for OOD detection. Importantly, we find this OOD behavior to be consistent with the corresponding finite-width networks. To overcome this limitation, useful function space properties can also be encoded in the prior in weight space, however, this can currently only be applied to a specified subset of the domain and thus does not inherently extend to OOD data. Finally, we argue that a trade-off between generalization and OOD capabilities might render the application of BNNs for OOD detection undesirable in practice. Overall, our study discloses fundamental problems when naively using BNNs for OOD detection and opens interesting avenues for future research.

## 1 Introduction

One of the challenges that the modern machine learning community is striving to tackle is the detection of unseen inputs for which predictions should not be trusted. This problem is also known as out-of-distribution (OOD) detection. The challenging nature of this task is partly rooted in the fact that there is no universal mathematical definition that characterizes an unseen input $\boldsymbol{x}^*$ as OOD as discussed in Sec. 2. Without such a definition, there is no foundation for deriving detection guarantees under verifiable assumptions. In this work, we consider OOD points as points that are unlikely under the distribution of the data $p(\boldsymbol{x})$ and points are outside the support of $p(\boldsymbol{x})$. Importantly, we do not claim to provide a meaningful definition of OOD, but rather argue that the justification of an OOD method can only be assessed theoretically once such a definition is provided. While it appears natural to tackle the OOD detection problem from a generative perspective by explicitly modelling $p(\boldsymbol{x})$, this paper is solely concerned with the question of how justified it is to deploy the predictive uncertainty of a Bayesian neural network (BNN) for OOD detection. As recent developments forecast an increasing integration of deep learning methods into industrial applications, it becomes essential to provide the theoretical groundings that justify the use of uncertainty for OOD detection, a task that is crucial for safety-critical applications of AI and reliable prediction-making.

When BNNs are used in supervised learning, the dataset is composed of inputs $\boldsymbol{x} \in \mathcal{X}$ and targets $\boldsymbol{y} \in \mathcal{Y}$ which are assumed to be generated according to some unknown process: $\mathcal{D} \overset{i.i.d.}{\sim} p(\boldsymbol{x})p(\boldsymbol{y} \mid \boldsymbol{x})$. The goal of learning is to infer from $\mathcal{D}$ alone the distribution $p(\boldsymbol{y} \mid \boldsymbol{x})$ in order to make predictions on unseen inputs $\boldsymbol{x}^*$. In the case of deep learning, this problem is approached by choosing a neural network $f(\cdot; \boldsymbol{w})$ parametrized by $\boldsymbol{w}$, and predictions are made via the conditional $p(\boldsymbol{y} \mid f(\boldsymbol{x}; \boldsymbol{w}))$ (also called the *likelihood* of $\boldsymbol{w}$ for given $\boldsymbol{x}, \boldsymbol{y}$). Assuming that the induced class of hypotheses contains some $\hat{\boldsymbol{w}}$ such that $p(\boldsymbol{y} \mid f(\boldsymbol{x}; \hat{\boldsymbol{w}})) = p(\boldsymbol{y} \mid \boldsymbol{x})$ almost everywhere on the support of $p(\boldsymbol{x})$, Bayesian statistics can be used to infer plausible models $p(\boldsymbol{w} \mid \mathcal{D})$ under the observed data given some prior knowledge $p(\boldsymbol{w})$ (see MacKay (2003)). This parametric description together with the network implicitly induces a prior over functions $p(\boldsymbol{f})$ (Williams, 1997; Fortuin, 2021). Believing in the validity of a subjective choice of prior (O'Hagan, 2004), Bayesian inference comes with a

multitude of benefits as it is less susceptible to overfitting, allows to incorporate new evidence without requiring access to past data (Farquhar & Gal, 2018) and provides interpretable uncertainties.

The uncertainty captured by a BNN can be coarsely categorized into *aleatoric* and *epistemic* uncertainty. Aleatoric uncertainty is irreducible and intrinsic to the data $p(\boldsymbol{y} \mid \boldsymbol{x})$. For instance, a blurry image might not contain enough information to identify unique object classes. On the other hand, a BNN also models epistemic uncertainty by maintaining a distribution over parameters $p(\boldsymbol{w} \mid \mathcal{D})$.[1] This distribution reflects the uncertainty about which hypothesis explains the data and can be reduced by observing more data. Arguably, aleatoric uncertainty is of little interest for detecting OOD inputs. However, the recent advent of overparametrized models which further extend the hypothesis class, deceptively motivated the idea that the epistemic uncertainty under a Bayesian framework might be intrinsically suitable to detect unfamiliar inputs, and

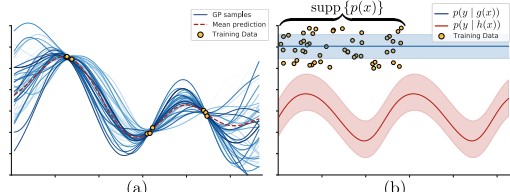

(a)  (b)

Figure 1: **(a)** GP regression with an RBF kernel illustrates uncertainty-based OOD detection. The prior variance over function values is only squeezed around training points, which leaves epistemic uncertainty high in OOD regions. **(b)** Conceptual illustration on why epistemic uncertainty is not necessarily linked to OOD detection (see main text for more details). Note, that the ground-truth $p(\boldsymbol{y} \mid \boldsymbol{x})$ is only defined within the support of $p(\boldsymbol{x})$.

therewith implying, that BNNs can be used for OOD detection. This conjecture is intuitively true if the following assumption holds: the hypotheses captured by $p(\boldsymbol{w} \mid \mathcal{D})$ must agree on their predictions for in-distribution samples but disagree for OOD samples (Fig. 1a). Certain Bayesian methods satisfy this assumption, e.g. a Gaussian process regression with an RBF kernel (cf. Sec. 4). However, the uncertainty induced by Bayesian inference does not in general give rise to OOD capabilities. This can easily be verified by considering the following thought experiment (Fig. 1b): Assume a model class with only two hypotheses $g(x)$ and $h(x)$ such that $g(x) \neq h(x) \, \forall x$ and data being generated according to $g(x)$ on a restricted domain. Once data is observed, we can commit to the ground-truth hypothesis $g(x)$ and thus lose epistemic uncertainty in- and out-of-distribution alike. Finite-width neural networks, by contrast, form a powerful class of models, and are often put into context with universal function approximators (Hornik, 1991). But is this fact in combination with Bayesian statistics enough to attribute them with good OOD capabilities? The literature often seems to imply that a BNN is intrinsically good at OOD detection. For instance, the use of OOD benchmarks when introducing new methods for approximate inference creates the false impression that the true posterior is a good OOD detector (Louizos & Welling, 2017; Pawlowski et al., 2017; Krueger et al., 2017; Henning et al., 2018; Maddox et al., 2019; Ciosek et al., 2020; D'Angelo & Fortuin, 2021). Our work is meant to start a discussion among researchers about the properties of OOD uncertainty of the exact Bayesian posterior of neural networks. To initiate this, we contribute as follows:

- We emphasize the importance of the prior in function space for OOD detection, which is induced by the choice of architecture (Sec. 4) and weight space prior (Sec. 5).

- We empirically show that exact inference in infinite-width networks under common architectural choices does not necessarily lead to desirable OOD behavior, and that these observations are consistent with results obtained via approximate inference in their finite-width counterparts.

- We furthermore study the OOD behavior in infinite-width networks by analysing the properties of the induced kernels. Moreover, we discuss desirable kernel features for OOD detection and show that also neural networks can approximately carry these features.

- We emphasize that the choice of weight-space prior has a strong effect on OOD performance, and that encoding desirable function space properties within unknown OOD domains into such prior is challenging.

- We argue that there is a trade-off between good generalization and having high uncertainty on OOD data. Indeed, improving generalization by incorporating prior knowledge (which is usually encoded in an input-domain agnostic manner) can negatively impact OOD uncertainties.

---

[1]Note, that the Bayesian treatment of network parameters does not account for all types of epistemic uncertainty such as the uncertainty stemming from model misspecification (Hüllermeier & Waegeman, 2021). We, however, assume the model to be correctly specified.

## 2 ON THE DIFFICULTY OF DEFINING OUT-OF-DISTRIBUTION INPUTS

While the intuitive notion of OOD sample points (or outliers) is commonly agreed upon, for instance as "an observation (or subset of observations) which appears to be inconsistent with the remainder of that set of data" (Barnett & Lewis, 1984), a mathematical formalization of the essence of an outlier is difficult (cf. SM A) and requires subjective characterizations (Zimek & Filzmoser, 2018). Often, methods for outlier detection are designed based on an intuitive notion (see Pimentel et al. (2014) for a review), and therewith only implicitly define a method-specific definition of OOD points. This also accounts for uncertainty-based OOD detection with neural networks, where a statistic of the predictive distribution that quantifies uncertainty (e.g., the entropy) is used to decide whether an input is considered OOD (Hendrycks & Gimpel, 2017). Commonly, the parameters $\boldsymbol{w}$ of a neural network are trained using an objective derived from $\mathbb{E}_{p(\boldsymbol{x})}\left[\mathrm{KL}\left(p(\boldsymbol{y}\mid\boldsymbol{x})||p(\boldsymbol{y}\mid f(\boldsymbol{x};\boldsymbol{w}))\right)\right]$ (i.e., loss functions based on the negative log-likelihood such as the cross-entropy or mean-squared error loss). Therefore, these networks only have calibrated uncertainties in-distribution with OOD uncertainties not being controlled for unless explicit training on OOD data is performed (eg., Hendrycks et al., 2019). The question this study is concerned with is therefore: *does the explicit treatment of parameter uncertainty via the posterior parameter distribution $p(\boldsymbol{w}\mid\mathcal{D})$ elicit provably high uncertainty OOD?*

Note, that we do not consider the problem of outlier detection within the training set (Zimek & Filzmoser, 2018), but rather ask whether a deployed model is able to *know what it does not know*.

## 3 BACKGROUND

In this section, we briefly introduce the concepts on which we base our argumentation in the coming sections. We start by introducing BNNs, which rely on approximate inference. We then discuss that in the non-parametric limit and under a certain choice of prior, a BNN converges to a Gaussian process, an alternative Bayesian inference framework where exact inference is possible. The connection between BNNs and Gaussian processes will later allow us to make interesting conjectures about OOD behavior. Finally, we introduce generalization bounds from the PAC-Bayes framework, which we will later use to argue that generalization and OOD detection can be conflicting objectives.

### 3.1 BAYESIAN NEURAL NETWORKS

In supervised deep learning, we typically construct a likelihood function from the conditional density $p(\boldsymbol{y}\mid f(\boldsymbol{x};\mathbf{w}))$, parameterized by a neural network $f(\boldsymbol{x};\boldsymbol{w})$, and the training data $\mathcal{D}=\{(\boldsymbol{x}_i,\boldsymbol{y}_i)\}_{i=1}^n$. In BNNs, this is used to form the posterior distribution of all likely network parameterizations: $p(\boldsymbol{w}\mid\mathcal{D})\propto\prod_{i=1}^n p(\boldsymbol{y}_i\mid f(\boldsymbol{x}_i;\boldsymbol{w}))\,p(\boldsymbol{w})$, where $p(\boldsymbol{w})$ is the prior distribution over weights. Crucially, when making a prediction with the Bayesian approach on a test point $\boldsymbol{x}^*$, we do not only use a single parameter configuration $\hat{\boldsymbol{w}}$ but we marginalize over the whole posterior, thus taking all possible explanations of the data into account: $p(\boldsymbol{y}^*\mid\boldsymbol{x}^*,\mathcal{D})=\int p(\boldsymbol{y}^*\mid f(\boldsymbol{x}^*;\boldsymbol{w}))\,p(\boldsymbol{w}\mid\mathcal{D})\,\mathrm{d}\boldsymbol{w}$.

### 3.2 GAUSSIAN PROCESS

Gaussian processes are established Bayesian machine learning models that, despite their strong scalability limitations, can offer a powerful inference framework. Formally (Rasmussen, 2004):

**Definition 1** *A Gaussian process (GP) is a collection of random variables, any finite number of which have a joint Gaussian distribution.*

Compared to parametric models, GPs have the advantage of performing inference directly in function space. A GP is defined by its mean $m(\boldsymbol{x})=\mathbb{E}\left[\boldsymbol{f}(\boldsymbol{x})\right]$ and covariance $C\left(\boldsymbol{f}(\boldsymbol{x}),\boldsymbol{f}(\boldsymbol{x}')\right)=\mathbb{E}\left[(\boldsymbol{f}(\boldsymbol{x})-m(\boldsymbol{x}))\left(\boldsymbol{f}(\boldsymbol{x}')-m(\boldsymbol{x}')\right)\right]$. The latter can be specified using a kernel function $k:\mathbb{R}^d\times\mathbb{R}^d\to\mathbb{R}$ and implies a prior distribution over functions:

$$p(\boldsymbol{f}|X)=\mathcal{N}\left(\mathbf{0},K(X,X)\right),\qquad(1)$$

where $K(X,X)_{ij}=k(\boldsymbol{x}_i,\boldsymbol{x}_j)$ is the kernel Gram matrix on the training inputs $X$, and $m(\boldsymbol{x})$ has been chosen to be 0. When observing the training data, the prior is reshaped to place more mass in the regions of functions that are more likely to have generated them, and this knowledge is then used to make predictions on unseen inputs $X_\star$. In probabilistic terms, this operation corresponds to conditioning the joint Gaussian prior: $p(\boldsymbol{f}_*|X_*,X,\boldsymbol{f})$. To model the data more realistically, we assume to not have direct access to the function values but to noisy observations: $\boldsymbol{y}=\boldsymbol{f}(\boldsymbol{x})+\epsilon$ with $\epsilon\sim\mathcal{N}(0,\sigma^2\mathbb{I}_d)$.

This assumption is formally equivalent to a Gaussian likelihood $p(\boldsymbol{y}|\boldsymbol{f}) = \mathcal{N}(\boldsymbol{y}|\boldsymbol{f}, \sigma^2)$.[2] The conditional distribution on the noisy observation can then be written as (Rasmussen, 2004):

$$
\begin{aligned}
p(\boldsymbol{f}_* \mid X_*, X, \boldsymbol{y}) &= \int d\boldsymbol{f} p(\boldsymbol{f}_* \mid X_*, X, \boldsymbol{f}) p(\boldsymbol{f} \mid X, \boldsymbol{y}) = \mathcal{N}\left(\bar{\boldsymbol{f}}_*, C(\boldsymbol{f}_*)\right) \\
\bar{\boldsymbol{f}}_* &= K(X_*, X)[K(X, X) + \sigma^2 \mathbb{I}]^{-1} \boldsymbol{y} \\
C(\boldsymbol{f}_*) &= K(X_*, X_*) - K(X_*, X)[K(X, X) + \sigma^2 \mathbb{I}]^{-1} K(X, X_*)
\end{aligned}
\tag{2}
$$

where $p(\boldsymbol{f} \mid X, \boldsymbol{y}) = \frac{p(\boldsymbol{y}|\boldsymbol{f})p(\boldsymbol{f}|X))}{p(\mathcal{D})}$ with $p(\mathcal{D})$ being the marginal likelihood.

**RBF kernel.** A commonly used kernel function for GPs is the squared exponential (RBF):

$$
k(\boldsymbol{x}, \boldsymbol{x}') = \exp\left(-\frac{1}{2l^2}\|\boldsymbol{x} - \boldsymbol{x}'\|_2^2\right) \quad ,
\tag{3}
$$

with the length scale $l$ being a hyperparameter. It is important to notice that the covariance between outputs is written exclusively as a function of the distance between inputs. As a consequence, points that are close in the Euclidean space have unitary covariance that decreases exponentially with the distance. As we will see in Sec. 4, this feature has important implications for OOD detection.

**The relation of infinite-width BNNs and GPs.** The connection between neural networks and GPs has recently gained significant attention. Neal (1996) showed that a 1-hidden layer BNN converges to a GP in the infinite-width limit. More recently, the work by Lee et al. (2018) and de G. Matthews et al. (2018) extended this result to deeper networks, called *neural network Gaussian processes* (NNGP). Crucially, the kernel function of the related GP strictly depends on the used activation function. To better understand this connection we consider a fully connected network with $L$ layers $l = 0, \ldots, L$ with width $H_l$. For each input we use $\mathbf{x}^l(\boldsymbol{x})$ to represent the post-activation with $\mathbf{x}^0 = \boldsymbol{x}$ and $\boldsymbol{f}^l$ the pre-activation so that $f_i^l(\boldsymbol{x}) = b_i^{l-1} + \sum_{j=1}^{H_{l-1}} w_{ij}^{l-1} x_j^{l-1}$ with $x_j^l = h(f_j^l)$ and $h(\cdot)$ a point-wise activation function. Furthermore, the weights and biases are distributed according to $b_j^l \sim \mathcal{N}(0, \sigma_b^2)$ and $w_{ij}^l \sim \mathcal{N}(0, \frac{\sigma_w^2}{H_l})$. Given the independence of the weights and biases it follows that the post-activations are independent as well. Hence, the central limit theorem can be applied, and for $H_{l-1} \to \infty$ we obtain $f^l(\boldsymbol{x}) \sim \mathcal{GP}(0, C^l)$. The covariance $C^l$ and therefore the prior over functions is specified by the kernel induced by the network architecture (Lee et al., 2018):

$$
\begin{aligned}
C^l(\boldsymbol{x}, \boldsymbol{x}') &= \overbrace{\mathbb{E}\left[f_i^l(\boldsymbol{x}) f_i^l(\boldsymbol{x}')\right]}^{k^l(\boldsymbol{x}, \boldsymbol{x}')} - \overbrace{\mathbb{E}[f_i^l(\boldsymbol{x})]\mathbb{E}[f_i^l(\boldsymbol{x})]}^{=0} \\
&= \sigma_b^2 + \sigma_w^2 \mathbb{E}_{(f_i^{l-1}(\boldsymbol{x}), f_i^{l-1}(\boldsymbol{x}')) \sim \mathcal{GP}(0, K^{l-1})}\left[h\left(f_i^{l-1}(\boldsymbol{x})\right), h\left(f_i^{l-1}(\boldsymbol{x}')\right)\right] \quad ,
\end{aligned}
\tag{4}
$$

with $K^l$ being the $2 \times 2$ covariance matrix formed by $k^l(\cdot, \cdot)$ using $\boldsymbol{x}$ and $\boldsymbol{x}'$. For the input layer, where no activation function is applied, the kernel is simply given by $k^0(\boldsymbol{x}, \boldsymbol{x}') = \sigma_b^2 + \sigma_w^2\left(\frac{\boldsymbol{x}^T \boldsymbol{x}'}{d}\right)$. For the remaining layers, instead, the kernel expression is determined by the non-linearity. Some activation functions like sigmoid or hyperbolic tangent do not admit a known analytical form and therefore require a Monte Carlo estimate of Eq. 4 (cf. Fig. S3). For others, Eq. 4 can be computed in closed form (e.g., Williams, 1997; Cho & Saul, 2009; Tsuchida et al., 2018; Pang et al., 2019; Pearce et al., 2020). We list some kernel functions that are important for this study below, such as the one for ReLU networks (Cho & Saul, 2009):

$$
\begin{aligned}
k_{\text{ReLU}}^l(\boldsymbol{x}, \boldsymbol{x}') &= \sigma_b^2 + \frac{\sigma_w^2}{2\pi}\sqrt{k^{l-1}(\boldsymbol{x}, \boldsymbol{x})k^{l-1}(\boldsymbol{x}', \boldsymbol{x}')}\left(\sin\theta_{\boldsymbol{x}, \boldsymbol{x}'}^{l-1} + (\pi - \theta_{\boldsymbol{x}, \boldsymbol{x}'}^{l-1})\cos\theta_{\boldsymbol{x}, \boldsymbol{x}'}^{l-1}\right) \\
\theta_{\boldsymbol{x}, \boldsymbol{x}'}^l &= \cos^{-1}\left(\frac{k^l(\boldsymbol{x}, \boldsymbol{x}')}{\sqrt{k^l(\boldsymbol{x}, \boldsymbol{x})k^l(\boldsymbol{x}', \boldsymbol{x}')}}\right)
\end{aligned}
\tag{5}
$$

Interestingly, for some non-linearities a kernel that carries similar properties as the RBF in Eq. 3 can be induced by neural networks. This is the case for the cosine activation function which has the

---

[2]Note, that the Gaussian likelihood assumption is commonly applied to neural network regression through the use of a mean-squared error loss.

following closed form solution in the single hidden layer case (Pearce et al., 2020):

$$k_{\cos}^1(\boldsymbol{x}, \boldsymbol{x}') = \sigma_b^2 + \frac{\sigma_w^2}{2}\left(\exp\left(-\frac{\sigma_w^2\|\boldsymbol{x} - \boldsymbol{x}'\|_2^2}{2d}\right) + \exp\left(-\frac{\sigma_w^2\|\boldsymbol{x} + \boldsymbol{x}'\|_2^2}{2d} - 2\sigma_b^2\right)\right) \quad (6)$$

Similarly, also the exponential activation function as deployed in RBF networks (Broomhead & Lowe, 1988) leads to similar properties. This particular class of neural networks defines computational units as linear combinations of radial basis functions $f(\boldsymbol{x}) = \sum_{j=1}^H w_j \exp\left(-\frac{1}{2\sigma_g^2}\|\boldsymbol{x} - \boldsymbol{\mu}_j\|^2\right) + b$. These networks, in the infinite-width case and when assuming a Gaussian prior over the centers $\boldsymbol{\mu}_j \sim \mathcal{N}(0, \sigma_\mu^2\mathbb{I})$ and $w_j \sim \mathcal{N}(0, \sigma_w^2)$, also converges to a Gaussian process with an analytical kernel function:

$$k_{\text{rbfnet}}^1(\boldsymbol{x}, \boldsymbol{x}') = \sigma_b^2 + \sigma_w^2\left(\frac{\sigma_e}{\sigma_\mu}\right)^d \exp\left(-\frac{\|\boldsymbol{x}\|^2}{2\sigma_m^2}\right)\exp\left(-\frac{\|\boldsymbol{x} - \boldsymbol{x}'\|^2}{2\sigma_s^2}\right)\exp\left(-\frac{\|\boldsymbol{x}'\|^2}{2\sigma_m^2}\right) \quad (7)$$

where $1/\sigma_e^2 = 2/\sigma_g^2 + 1/\sigma_\mu^2$, $\sigma_s^2 = 2\sigma_g^2 + \sigma_g^4/\sigma_\mu^2$ and $\sigma_m^2 = 2\sigma_\mu^2 + \sigma_g^2$. The equivalence with the RBF kernel is explicit in the limit $\sigma_\mu^2 \to \infty$ (Williams, 1997).

### 3.3 PAC-Bayes generalization bound

Considering the supervised learning framework introduced in Sec. 1, the PAC theory (Valiant, 1984) establishes a probabilistic bound on the generalization error of a given predictor. The theory is referred to as PAC-Bayes (McAllester, 1999; Catoni, 2007) when a prior distribution $p$ is defined to incorporate prior domain knowledge. Considering a distribution[3] $q$ on the hypothesis $h \in \mathcal{H}$ and the space $\mathcal{B}(\mathcal{Y})$ of all conditional distributions $p(\boldsymbol{y} \mid h(\boldsymbol{x}))$, we can define a bounded loss function such that $l : \mathcal{B}(\mathcal{Y}) \times \mathcal{Y} \to [0, 1]$. This gives rise to the empirical and true risk as $R_\mathcal{D}(q) := \frac{1}{N}\sum_{i=1}^N \mathbb{E}_{h \sim q}[l(p(\boldsymbol{y} \mid h(\boldsymbol{x}_i)), \boldsymbol{y}_i)]$ and $R(q) := \mathbb{E}_{(\boldsymbol{x}_\star, \boldsymbol{y}_\star) \sim p(\boldsymbol{x}, \boldsymbol{y})}\mathbb{E}_{h \sim q}[l(p(\boldsymbol{y} \mid h(\boldsymbol{x}_\star)), \boldsymbol{y}_\star)]$, respectively, where $p(\boldsymbol{x}, \boldsymbol{y}) = p(\boldsymbol{x})p(\boldsymbol{y} \mid \boldsymbol{x})$. Note that when $q$ is the Bayesian posterior as, for instance, in the GP case, the Bayes risk can be defined as: $R_B(q) := \mathbb{E}_{(\boldsymbol{x}_\star, \boldsymbol{y}_\star) \sim p(\boldsymbol{x}, \boldsymbol{y})}[l(\mathbb{E}_{h \sim q}[p(\boldsymbol{y} \mid h(\boldsymbol{x}_\star))], \boldsymbol{y}_\star)]$ and it can be shown that $R_B(q) \leq 2R(q)$ (Seeger, 2003). Therefore, a bound over $R$ also implies a bound over $R_B$ (Seeger, 2003). The PAC bound gives a probabilistic upper bound on the true risk $R(q)$ in terms of the empirical risk $R_\mathcal{D}(q)$ for a training set $\mathcal{D}$ as formulated in the following theorem:

**Theorem 1 (PAC-Bayes theorem (Catoni, 2007))** *For any distribution $p(\boldsymbol{x}, \boldsymbol{y})$ over $\mathcal{X} \times \mathcal{Y}$, for any distribution $q$ and prior $p$ on a hypothesis space $\mathcal{H}$, for any $\delta \in (0, 1]$ and $\beta > 0$ the following holds with probability at least $1 - \delta$ over the training set $\mathcal{D} \sim p(\boldsymbol{x}, \boldsymbol{y})$ of cardinality $N$:*

$$\forall q : R(q) \leq \frac{1}{1 - e^{-\beta}}\left[1 - \exp\left(-\beta R_\mathcal{D}(q) - \frac{1}{N}\left(KL(q\|p) + \log\frac{1}{\delta}\right)\right)\right]. \quad (8)$$

For a fixed $p$, $\mathcal{D}$, $\delta$, minimizing Eq. 8 is equivalent to minimizing $NR_\mathcal{D}(q) + \text{KL}(q\|p)$. The requirement of a bounded loss function makes the use of the mean-squared error, and thus the negative log-likelihood, not applicable to this bound. Nevertheless, as long as the loss function measures the quality of predictions, the bound in Eq. 8 can be used to assess the generalization capabilities of a model. For this purpose, in our analysis we use as a surrogate loss the following Reeb et al. (2018): $l_{\exp}(p(\boldsymbol{y} \mid h), \boldsymbol{y}) = 1 - \exp\left[-\frac{(\mathbb{E}_{\boldsymbol{y}}[p(\boldsymbol{y}|h)] - y)^2}{\sigma^2}\right]$. Interestingly, for small deviations we can take the first-order Taylor expansion and recover the MSE $l_{\exp}(\mathbb{E}_{\boldsymbol{y}}[p(\boldsymbol{y} \mid h)], \boldsymbol{y},) \approx (\mathbb{E}_{\boldsymbol{y}}[p(\boldsymbol{y} \mid h)] - \boldsymbol{y})^2/\sigma^2$.

## 4 THE ARCHITECTURE STRONGLY INFLUENCES OOD UNCERTAINTIES

In the previous section, we recalled the connection between BNNs and GPs, namely that Bayesian inference in an infinite-width neural network can be studied in the GP framework (assuming a proper choice of weight-space prior). This connection allows studying how the kernels induced by architectural choices shape the prior in function space, and how these choices ultimately determine OOD behavior. In this section, we analyze this OOD behavior for traditional as well as NNGP-induced kernels. To minimize the impact of the approximations on our results, we exclusively focus on conjugate settings, i.e., regression (see SM C.1 for classification results). Furthermore, this choice

---

[3]Note that this distribution does not necessarily need to be the Bayesian posterior.

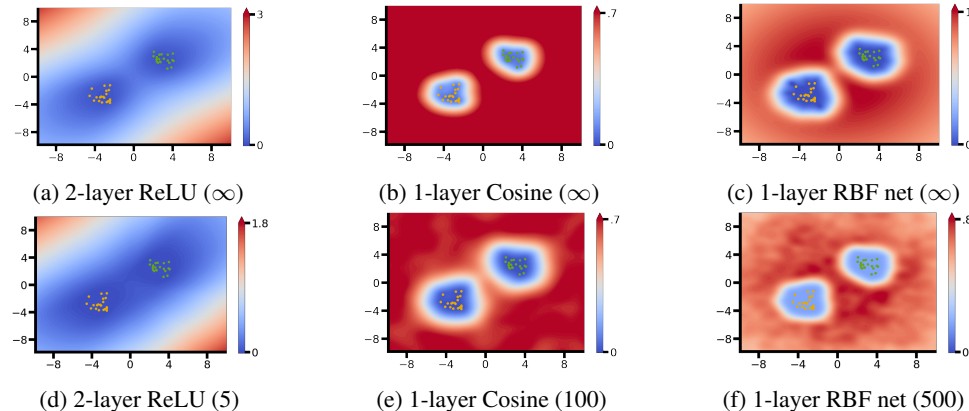

(a) 2-layer ReLU ($\infty$)  (b) 1-layer Cosine ($\infty$)  (c) 1-layer RBF net ($\infty$)

(d) 2-layer ReLU (5)  (e) 1-layer Cosine (100)  (f) 1-layer RBF net (500)

Figure 2: **Standard deviation $\sigma(\boldsymbol{f}_*)$ of the predictive posterior of BNNs.** We perform Bayesian inference on a mixture of two Gaussians dataset considering different priors in function space induced by different architectural choices. The problem is treated as regression task to allow exact inference in combination with GPs (a, b, c). Predictive uncertainties for finite-width networks are obtained using HMC (d, e, f).

of Gaussian likelihood induces a direct correspondence between function values and outcomes up to noise corruptions. Hence, prior knowledge about outcomes can be encoded in function space through the choice of a meaningful function space prior.

**Uncertainty quantification for OOD detection.** Uncertainty can be quantified in multiple ways, but is often measured as the entropy of the predictive posterior. The predictive posterior, however, captures both aleatoric and epistemic uncertainty, which does not allow a distinction between OOD and ambiguous inputs (Mukhoti et al., 2021). While our choice of likelihood does not permit the modelling of input-dependent uncertainty (Gaussian with fixed variance), a softmax classifier can capture heteroscedastic aleatoric uncertainty arbitrarily well by outputting an input-dependent categorical distribution.[4] For this reason, uncertainty should be quantified in a way that allows OOD detection to be based on epistemic uncertainty only. The function space view of GPs naturally provides such measure of epistemic uncertainty by considering the standard deviation of the posterior over function values $\sigma(\boldsymbol{f}_*) \equiv \sqrt{\operatorname{diag}\left(C(\boldsymbol{f}_*)\right)}$ (illustrated in Fig. 1a). Such measure can be naturally translated to BNNs by looking at the disagreement between network outputs when sampling from $p(\boldsymbol{w} \mid \mathcal{D})$. Note that in classification tasks all models drawn from $p(\boldsymbol{w} \mid \mathcal{D})$ might lead to a high-entropy softmax without disagreement, and therefore only an uncertainty measure based on model disagreement prevents one from misjudging ambiguous points as OOD. As OOD detection is the focus of this paper, we always quantify uncertainty in terms of model disagreement.

**Bayesian statistics and OOD detection have no intrinsic connection.** As conceptually illustrated in Fig. 1b and demonstrated using exact Bayesian inference in Fig. 3b, predictive uncertainties are not necessarily reflective of $p(\boldsymbol{x})$ (and thus not suited for OOD detection), irrespective of how well the underlying task is solved.

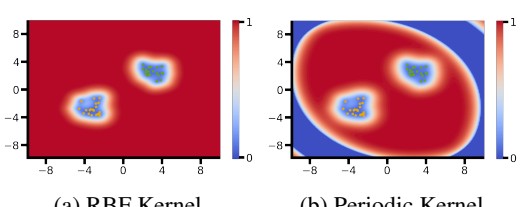

(a) RBF Kernel  (b) Periodic Kernel

Figure 3: **Standard deviation $\sigma(\boldsymbol{f}_*)$ of the predictive posterior using GPs with common kernel functions.** GP regression is performed on the same dataset as in Fig. 2.

**GP regression with an RBF kernel.** We next examine the predictive posterior of a GP with squared exponential (RBF) kernel (Eq. 3). Fig. 3a shows epistemic uncertainty as the standard deviation of $p(\boldsymbol{f}_* \mid X_*, X, \boldsymbol{y})$. We can notice that $\sigma(\boldsymbol{f}_*)$ nicely captures the data manifold and is thus well suited for OOD detection. This behavior can be understood by considering Eq. 2 while noting that $k(\boldsymbol{x}, \boldsymbol{x}') = \text{const}$ if $\boldsymbol{x} = \boldsymbol{x}'$, and that the

---

[4]There are also expressive likelihood choices to capture heteroscedastic uncertainty for continuous variables (e.g., Zięba et al., 2020).

variance of posterior function values can be written as $\sigma^2(\boldsymbol{f}_*) = k(\boldsymbol{x}_*, \boldsymbol{x}_*) - \sum_{i=1}^{n} \beta_i(\boldsymbol{x}_*) k(\boldsymbol{x}_*, \boldsymbol{x}_i)$ (cf. SM B), where $\beta_i$ are dataset- and input-dependent. The second term is reminiscent of using kernel density estimation (KDE) to approximate $p(\boldsymbol{x})$, applying a Gaussian kernel, while the first term is the (constant) prior variance. Hence, the link between Bayesian inference and OOD detection can be made explicit, as the posterior variance is inversely related to the input distribution.

Loosely speaking, in GP regression as in Eq. 2 and kernels where the KDE analogy holds, epistemic uncertainty can roughly be described as $\text{const} - p(\boldsymbol{x})$.[5] In this view, one starts with high (prior) uncertainty everywhere, which is only reduced where data is observed. By contrast, learning a normalized generative model (e.g., using normalizing flows (Papamakarios et al., 2019)) often requires to start from an arbitrary probability distribution.

**The OOD behavior induced by NNGP kernels.** Fig. 2a illustrates $\sigma(\boldsymbol{f}_*)$ for an infinite-width 2-layer ReLU network (see Fig. S2 for other common architectural choices). It is already visually apparent, that in this case the kernel is less suited for OOD detection compared to an RBF kernel. Moreover, we cannot justify why OOD detection based on $\sigma(\boldsymbol{f}_*)$ would be principled for this kernel as the KDE analogy does not hold. This is due to two reasons: (1) the prior uncertainty $k(x_*, x_*)$ is not constant (Fig. S1), and (2) we empirically do not observe that $k(\boldsymbol{x}, \boldsymbol{x}')$ can be related to a distance measure (Fig. S4). We therefore argue that more theoretical work is necessary if one aims to justify uncertainty-based OOD detection with common architectural choices from the perspective of NNGP kernels. On this note, maintaining parameter uncertainty and being able to detect OOD samples are often considered crucial requirements of systems deployed in safety-critical applications. Given that Bayesian inference is not intrinsically linked to OOD detection, care should be taken to precisely communicate safety-relevant capabilities of BNNs to practitioners.

However, the NNGP perspective also allows us to choose an architecture such that the BNN's uncertainty resembles the, for these problems, desirable OOD behavior of the GP with RBF kernel described above. In particular, the cosine (Eq. 6) and RBF (Eq. 7) network induce kernels that are related to the RBF kernel. The last term in Eq. 6 quickly converges to zero for moderate norms of $\boldsymbol{x}$ or $\boldsymbol{x}'$ (or high $\sigma_b^2$), which explains why the uncertainty behavior of Fig. 2b is qualitatively identical to Fig. 3a. In case of the RBF network, the RBF kernel is recovered if $\|\boldsymbol{x}\|$ and $\|\boldsymbol{x}'\|$ are small compared to $\sigma_m^2$, explaining why the uncertainty faints towards the boundaries of Fig. 2c.

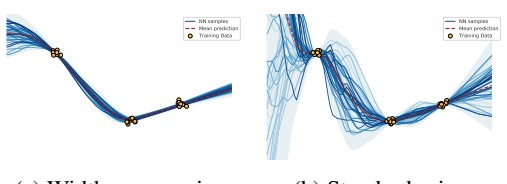

(a) Width-aware prior      (b) Standard prior

Figure 4: **The importance of the choice of weight prior** $p(\boldsymbol{w})$**.** Here, we perform 1d regression using HMC with either **(a)** a width-aware prior $\mathcal{N}(0, \frac{\sigma_w^2}{H_l})$ or **(b)** a standard prior $\mathcal{N}(0, \sigma_w^2)$.

These examples show that in the non-parametric limit and for (low-dimensional) regression tasks, BNNs can be constructed such that uncertainty-based OOD detection can be justified through mathematical argumentation. We next study whether the observations made in the infinite-width limit are relevant for studying finite-width neural networks.

**Infinite-width uncertainty is consistent with the finite-width uncertainty.** For finite-width BNNs exact Bayesian inference is intractable. To mitigate the effects of approximate inference we resort to Hamiltonian Monte Carlo (HMC, Duane et al., 1987; Neal et al., 2011). Fig. 2d to 2f show an estimate of $\sigma(\boldsymbol{f}_*)$ for finite-width networks corresponding to the non-parametric limits studied above (illustrations with another dataset can be found in Fig. S5). Already for moderate layer widths, the modelled uncertainty resembles the one of the corresponding NNGP. Given this close correspondence, we conjecture that the tools available for the infinite-width case are useful for designing architectural guidelines that enhance OOD detection. We illustrated this on low-dimensional problems by studying desirable function space properties induced by the RBF kernel, which can be translated to BNN architectures.

## 5 THE CHOICE OF WEIGHT SPACE PRIOR MATTERS FOR OOD DETECTION

For a neural network, the prior in function space is induced by the architecture and the prior in weight space (Wilson & Izmailov, 2020).

---

[5]Note, the KDE approximation of $p(\boldsymbol{x})$ is likely to deteriorate massively if the dimensionality of $\boldsymbol{x}$ increases.

In the previous section, we restricted ourselves to a particular class of weight space priors which allowed us to study the architectural choices in the infinite-width limit. While, in practice, a wide variety of weight space prior choices might lead to good generalization (with respect to test data from $p(\boldsymbol{x})p(\boldsymbol{y} \mid \boldsymbol{x})$, e.g., cf. Izmailov et al. (2021b)), the uncertainty behavior that is induced OOD might vary drastically. This is illustrated in Fig. 4, where for the same network two different choices of $p(\boldsymbol{w})$ lead to vastly different predictive uncertainties despite the fact that both choices fit the data well.

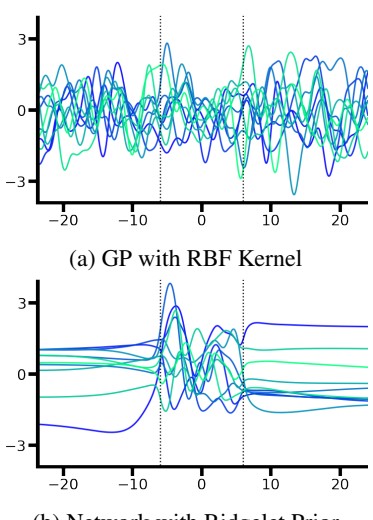

(a) GP with RBF Kernel

(b) Network with Ridgelet Prior

In Sec. 4, we use the infinite-width limit to obtain a function space view that allows to make interesting conjectures about predictive uncertainties on OOD data. Recently, multiple studies suggested ways to either explicitly encode function space properties into the weight space prior $p(\boldsymbol{w})$ or to use a function space prior when performing approximate inference in neural networks (Flam-Shepherd et al., 2017; Sun et al., 2019; Tran et al., 2020; Matsubara et al., 2021). However, all these methods depend on the specification of a set $\mathcal{X}_R \subseteq \mathcal{X}$ on which desired function space properties will be enforced (Fig. 5). For instance, the Ridgelet prior proposed by Matsubara et al. (2021) provides an asymptotically correct weight space prior construction that induces a given GP prior. Thus, on $\mathcal{X}_R$ this method allows to meaningfully encode prior knowledge to guide Bayesian inference. But, an a priori specification of a set $\mathcal{X}_R$ which can be largely covered by samples $\boldsymbol{x}_R \in \mathcal{X}_R$ to ensure the desired prior specification on $\mathcal{X}_R$ is practically challenging, and conceptually related to the idea of training on OOD data to calibrate respective uncertainties (cf. Sec. 2). Note, we do not aim to phrase this OOD effect as a drawback of these methods, as we do not consider Bayesian inference to be intrinsically related to OOD detectors. It is, however, important to keep in mind that function space properties deemed beneficial for OOD detection are not straightforward to induce when working in weight space.

Figure 5: **OOD challenges when encoding function space properties into weight space prior.** **(a)** Samples from a GP prior $p(\boldsymbol{f}|X)$. **(b)** Samples from a 1-layer Tanh network (width: 3000) using the Ridgelet prior (Matsubara et al., 2021) corresponding to the GP in (a). Dotted lines denote the domain $\mathcal{X}_R$ within which the Ridgelet prior was matched to the target GP.

## 6 A TRADE-OFF BETWEEN GENERALIZATION AND OOD DETECTION

An important desideratum that modelers attempt to achieve when applying Bayesian statistics is good generalization through the incorporation of relevant prior knowledge. Is this desideratum generally in conflict with having high uncertainty on OOD data? In this section, we provide arguments indicating that the answer to this question can be *yes*. Consider data with known periodic structure inside the support of $p(\boldsymbol{x})$ (Fig. 6).

Table 1: Empirical risk, KL divergence, PAC bound ($\delta = 0.01$) and log marginal likelihood for the examples shown in Fig. 6.

|  | $R_{\mathcal{D}}(q)$ | $\mathrm{KL}(q, p)$ | PAC | $\log p(\mathcal{D})$ |
|---|---|---|---|---|
| **RBF** | $0.117^{\pm.008}$ | $8.336^{\pm.191}$ | $0.563^{\pm.001}$ | $-31.536^{\pm.756}$ |
| **ESS** | $0.102^{\pm.008}$ | $7.231^{\pm.093}$ | $0.519^{\pm.011}$ | $-27.135^{\pm.706}$ |

We can either choose to ignore our prior knowledge by selecting a GP prior with RBF kernel (Fig. 6a), or we explicitly incorporate domain knowledge using a function space prior that allocates its mass on periodic functions e.g. the exp-sine-squared (ESS) kernel.[6] In the former case, we know that OOD uncertainties will be useful for OOD detection (Sec. 4). In the latter case, however, we see that uncertainties do not reflect $p(\boldsymbol{x})$. By contrast, the roles are reversed when it comes to assessing generalization.

---

[6]Note, that such incorporation of prior knowledge is often done in a way that is agnostic to the unknown support of $p(\boldsymbol{x})$.

Here, the choice of a periodic kernel has clear benefits, as we can see visually and by comparing the generalization bound introduced in Sec. 3.3 which can be tightened by minimizing $NR_{\mathcal{D}}\left(p(\boldsymbol{f} \mid \mathcal{D})\right) +$ KL $\left(p(\boldsymbol{f} \mid \mathcal{D})||p(\boldsymbol{f})\right)$. As reported in Table 1, the bound's value for the ESS kernel is lower than the RBF. Interestingly, the bound is minimized if both terms are minimized, and the second term explicitly asks the posterior to remain close to the prior. Therefore,

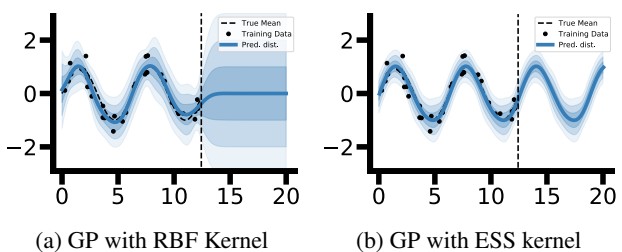

| (a) GP with RBF Kernel | (b) GP with ESS kernel |

Figure 6: **Generalization and OOD detection.** Mean and the first three standard deviations of the predictive posterior for different function space priors.

the second term benefits from restricting the prior function space through the incorporation of prior knowledge, counteracting the need of a rich function space for inducing high OOD uncertainties (Fig. 1). Consistent with this, also the log marginal likelihood $\log p(\mathcal{D})$ in Table 1 is higher for the ESS kernel. An orthogonal problem often considered in the literature is generalization under dataset (or covariate) shift (Quiñonero-Candela et al., 2009; Snoek et al., 2019; Izmailov et al., 2021a). In this case, one deliberately seeks to provide meaningful predictions on test data $p_{\text{test}}(\boldsymbol{x})$ that might not overlap in support with our training input distribution $p(\boldsymbol{x})$. Therefore, prior knowledge needs to explicitly encode how to obtain "generalization on OOD data" as the data cannot speak for themselves. Such priors are proposed in Izmailov et al. (2021a), but also Fig. 6b can be viewed as an example of how prior encoding specifies how to generalize to OOD data.

## 7    ON THE PRACTICAL VALIDATION OF OOD PROPERTIES

Relating the epistemic uncertainty induced by a BNN to $p(\boldsymbol{x})$ opens up interesting new possibilities. As we show in SM B, epistemic uncertainty of a GP with RBF kernel can be related to a KDE approximation of $p(\boldsymbol{x})$. Therefore, one may consider the epistemic uncertainty as an energy function to create a generative model from which to sample from the input space (Fig. 7)

.Moreover, this approach may be used to empirically validate the OOD capabilities of a BNN. Indeed, OOD performance is commonly validated by selecting a specific set of known OOD datasets (Izmailov et al., 2021b). However, as the OOD space comprises everything except the in-distribution data, it is infeasible to gain good coverage on high-dimensional data with such testing approach. We therefore suggest a reverse approach that checks the consistency in regions of certainty instead of uncertain ones by sampling via the epistemic uncertainty. Indeed, if these samples and thus the generative model based on uncertainty estimates are consistent with the in-

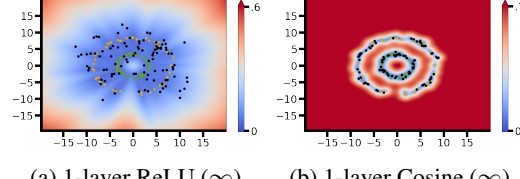

| (a) 1-layer ReLU ($\infty$) | (b) 1-layer Cosine ($\infty$) |

Figure 7: **Sampling from regions of low uncertainty.** Here, we treat epistemic uncertainty (measured as $\sigma(\boldsymbol{f}_*)$) as an energy function and perform rejection sampling (black dots). If uncertainty faithfully captures $p(\boldsymbol{x})$, these samples should resemble in-distribution data.

distribution data then a strong indication for trustworthiness is provided. Additionally, a discrepancy measure (Liu et al., 2016; Gretton et al., 2012) can be used to compare the generated samples and training data points and quantitatively assess to which extent the epistemic uncertainty is related to $p(x)$. Moreover, this approach can open up new research opportunities. For instance, a continual learner may use its uncertainty to generate its own replay data to combat forgetting (cf. SM C.3).

## 8    CONCLUSION

We challenge the common view that uncertainty-based OOD detection with BNNs is intrinsically justified. Our arguments are all based on low-dimensional problems and cannot easily be generalized to real-world problems, but our observations are consistent with the fact that empirically BNNs are often only marginally superior in OOD detection compared to models that do not maintain epistemic uncertainty (Snoek et al., 2019; Henning et al., 2021). Overall, this work highlights fundamental limitations of BNNs for OOD detection that are not solely explained by the use of approximate inference.

**Ethics Statement.** When machine learning algorithms enter real-world applications, they are not anymore embedded in a controlled environment. Instead, they have to *know what they don't know*, which encloses the notion of OOD detection. It is therefore of utmost importance to discuss and understand the justifications for employing uncertainty-based OOD detection. The purpose of our paper is to disclose in a simple but clear manner that these justifications are not obvious, and thus calls for more research in this direction. Our illustrative arguments should, however, not be misinterpreted or generalized, for instance, by assuming OOD properties we derive from an RBF kernel can be applied to high-dimensional image manifolds.

**Reproducibility Statement.** The source code to reproduce all experiments is available at: *anonymous link*.

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

# SUPPLEMENTARY MATERIAL: UNCERTAINTY-BASED OUT-OF-DISTRIBUTION DETECTION REQUIRES SUITABLE FUNCTION SPACE PRIORS

**Anonymous authors**

## A WHAT IS AN OUT-OF-DISTRIBUTION INPUT?

Pimentel et al. (2014) reviews methods for outlier detection, putting them coarsely into five categories: (1) probabilistic, (2) distance-based, (3) reconstruction-based, (4) domain-based, and (5) information-theoretic. Our focus lies on a probabilistic characterization of an OOD point, where a statistical criterion allows to decide whether a given input is significantly different from the observed training population. In this section, we discuss pitfalls regarding obvious choices for such a criterion in order to highlight the difficulties that arise when attempting to agree on a single (or application-dependent) mathematical definition of outliers.

There are many possible definitions that could be considered for a point to be OOD as we will outline below. Any of these definitions may change the notion of OOD and will therefore affect how a BNN should be designed such that predictive uncertainty adheres to the underlying OOD definition. Considering a generative process $p(\boldsymbol{x})$, the first question arising is regarding the regions outside the support of $p(\boldsymbol{x})$ or even the space outside the manifold where samples $\boldsymbol{x}$ are defined on. For instance, assume the data to be images embedded on a lower-dimensional manifold. Are points outside this manifold clearly OOD, given that minor noise corruptions are likely to leave the manifold? Even disregarding these topological issues solely focusing on the density $p(\boldsymbol{x})$, makes the distinction between in- and out-of-distribution challenging. For instance, a threshold-criterion on the density might cause samples in a zero probability region to be considered in-distribution (Nalisnick et al., 2019a). To overcome such challenges, one may resort to concepts from information theory, such as the notion of a typical set (MacKay, 2003; Nalisnick et al., 2019b). Unfortunately, an OOD criterion based on this notion would require looking at sets rather than individual points. We hope that this short outline highlights the challenges regarding the definition of OOD, but also clarifies that a proper definition is relevant when assessing OOD capabilities on high-dimensional data, where a visual assessment as in Fig. 2 is not possible.

## B ON THE RELATION BETWEEN GP REGRESSION AND KERNEL DENSITY ESTIMATION

GP regression with an RBF kernel allows a direct understanding of the OOD capabilities that a Bayesian posterior may possess, as the a priori uniform epistemic uncertainty is reduced in direct correspondence to density of $p(\boldsymbol{x})$.

Below, we rewrite the variance of an input $\boldsymbol{x}_*$ as defined in Eq. 2:

$$\sigma^2(\boldsymbol{f}_*) = k(\boldsymbol{x}_*, \boldsymbol{x}_*) - \sum_{i=1}^{n} \beta_i(\boldsymbol{x}_*) k(\boldsymbol{x}_*, \boldsymbol{x}_i) \tag{9}$$

with $\beta_i(\boldsymbol{x}_*) = \sum_{j=1}^{n} \left( K(X, X) + \sigma^2 \mathbb{I} \right)_{ij}^{-1} k(\boldsymbol{x}_*, \boldsymbol{x}_j)$.

Note, that $k(\boldsymbol{x}_*, \boldsymbol{x}_*)$ is a positive constant for an RBF kernel, and that $\beta_i(\boldsymbol{x}_*) \geq 0$. Furthermore, as the RBF kernel $k(\boldsymbol{x}_*, \boldsymbol{x}_j)$ behaves exponentially inverse to the distance between $\boldsymbol{x}_*$ and $\boldsymbol{x}_j$, $\beta_i(\boldsymbol{x}_*)$ is approximately zero for all $\boldsymbol{x}_*$ that are far from all training points. In addition, a training point $\boldsymbol{x}_i$ can only decrease the prior variance if it is close to $\boldsymbol{x}_*$. This analogy closely resembles the philosophy of KDE, which becomes an exact generative model in the limit of infinite data and bandwidth $l \to 0$.

## C ADDITIONAL EXPERIMENTS AND RESULTS

In this section, we report additional experiments and results.

Fig. S1 shows the prior's standard deviation over function values $\sqrt{k(\boldsymbol{x}_*, \boldsymbol{x}_*)}$ for several NNGP kernels. Note, a kernel that a priori treats locations $x_*$ differently might not be desirable for OOD detection as the data's influence on posterior uncertainties might be hard to interpret (cf. Sec. 4).

Fig. S2 shows GP regression results with the GMM dataset (Sec. D) for several NNGP kernels. The plots show that the underlying task can be solved well with all considered architectures (as indicated by the predictive mean $\bar{\boldsymbol{f}}_*$ that captures the ground-truth targets in-distribution), even though the uncertainty behavior OOD is vastly different and not reflective of $p(\boldsymbol{x})$.

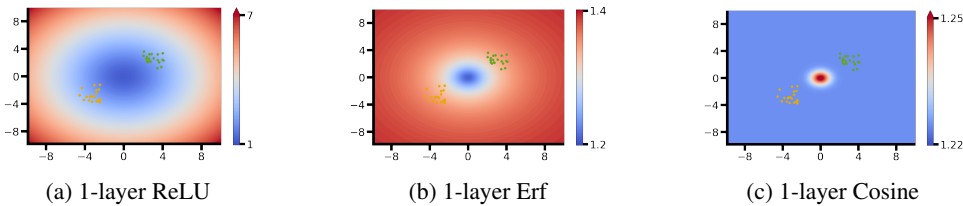

(a) 1-layer ReLU        (b) 1-layer Erf        (c) 1-layer Cosine

Figure S1: **NNGP kernel values $\sqrt{k(\boldsymbol{x}_*, \boldsymbol{x}_*)}$ for various architectural choices.** Note, that the NNGP kernel value $k(\boldsymbol{x}_*, \boldsymbol{x}_*)$ represents the prior variance of function values under the induced GP prior at the location $x_*$ (cf. Eq. 1). As emphasized in Sec. 4, $k(\boldsymbol{x}_*, \boldsymbol{x}_*)$ is constant for an RBF kernel, which has important implications for OOD detection, such as that a priori (before seeing any data) all points are treated equally. This is not the case for the ReLU kernel, which has an angular dependence and depends on the input norm (cf. Eq. 5 and its dependence on $k^0(\boldsymbol{x}, \boldsymbol{x}')$). The kernel induced by networks using an error function (Erf) or a cosine as nonlinearity seems to be more desirable in this respect (note the scale of the colorbars).

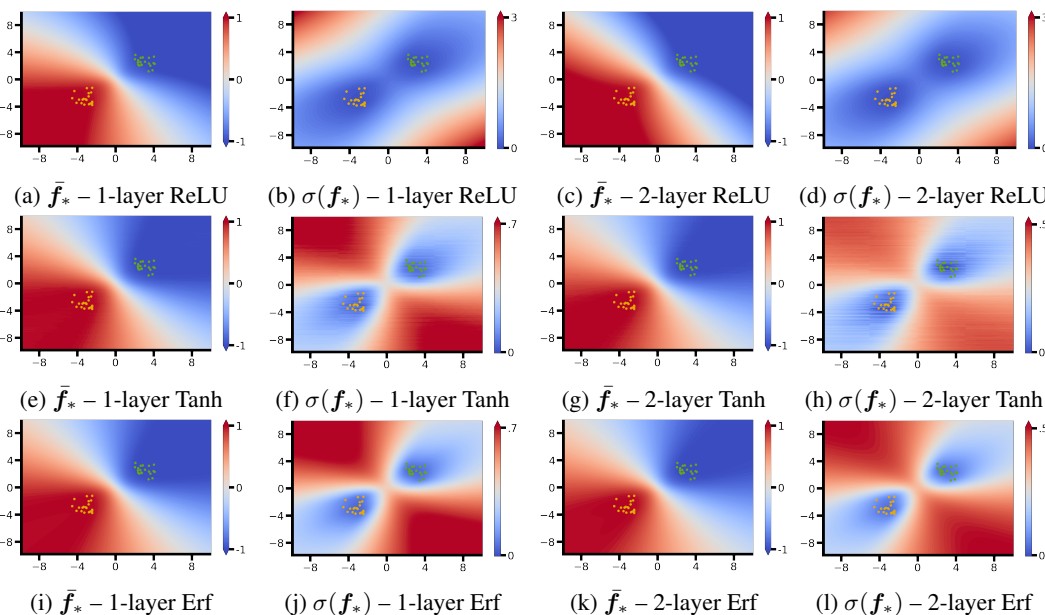

(a) $\bar{\boldsymbol{f}}_*$ – 1-layer ReLU   (b) $\sigma(\boldsymbol{f}_*)$ – 1-layer ReLU   (c) $\bar{\boldsymbol{f}}_*$ – 2-layer ReLU   (d) $\sigma(\boldsymbol{f}_*)$ – 2-layer ReLU

(e) $\bar{\boldsymbol{f}}_*$ – 1-layer Tanh   (f) $\sigma(\boldsymbol{f}_*)$ – 1-layer Tanh   (g) $\bar{\boldsymbol{f}}_*$ – 2-layer Tanh   (h) $\sigma(\boldsymbol{f}_*)$ – 2-layer Tanh

(i) $\bar{\boldsymbol{f}}_*$ – 1-layer Erf   (j) $\sigma(\boldsymbol{f}_*)$ – 1-layer Erf   (k) $\bar{\boldsymbol{f}}_*$ – 2-layer Erf   (l) $\sigma(\boldsymbol{f}_*)$ – 2-layer Erf

Figure S2: **Mean $\bar{\boldsymbol{f}}_*$ and standard deviation $\sigma(\boldsymbol{f}_*)$ of the posterior $p(\boldsymbol{f}_* \mid X_*, X, \boldsymbol{y})$ for GP regression with NNGP kernels** for various architectural choices, such as number of layers or nonlinearity. Note, that Tanh and Erf nonlinearities are quite similar in shape, which is reflected in the similar predictive posterior that is induced by these networks. We use the analytically known kernel expression for the Erf kernel (Williams, 1997), and use MC sampling for the Tanh network (cf. Fig. S3).

Fig. S3 (in combination with Fig. S2) highlights that also approximated kernel values can be used to study architectures with no known closed-form solution for Eq. 4, as the posterior seems to be only marginally effected by the MC estimation.

Fig. S4 visualizes that NNGP kernels for common architectural choices do not encode for Euclidean distances. Kernels that monotonically decrease with increasing distance are, however, important to apply the KDE interpretation of SM B.

Finally, in Fig. S5 we investigate a more challenging dataset with two concentric rings. Note, that the center region as well as the region between the two rings can be considered OOD (not included in the support of $p(\boldsymbol{x})$). A good uncertainty-based OOD detector should therefore depict high uncertainties in those regions, which is not the case for ReLU networks.

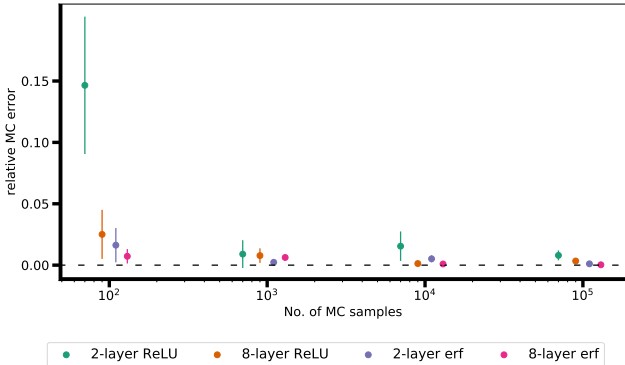

Figure S3: **Monte Carlo error when estimating NNGP kernel values.** Eq. 4 requires estimation whenever no analytic kernel expression is available (for instance, when using a hyberbolic tangent non-linearity as in Fig. S2). Here, we visualize the error caused by this approximation for ReLU and error function (erf) networks when computing kernel values $k(\boldsymbol{x}_*, \boldsymbol{x}_*)$. Eq. 4 requires a recursive estimation of expected values, where we estimate each of them using $N$ samples ($N \in [10^2, 10^3, 10^4, 10^5]$). Note, that even small errors can cause eigenvalues of the kernel matrix to become negative. However, with our chosen likelihood variance of $\sigma = 0.02$ we experience no numerical instabilities during inference, and obtain consistent results using either analytic or estimated ($N = 10^5$) kernel matrices.

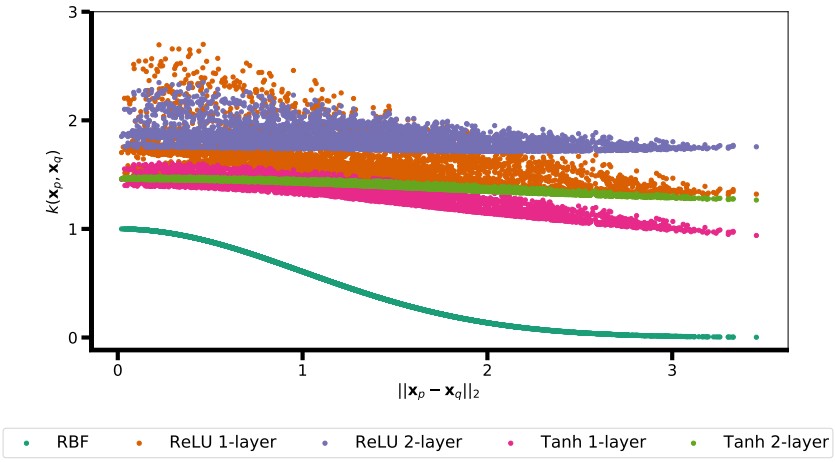

Figure S4: **NNGP kernel values do not generally reflect Euclidean distances.** This figure shows kernel values $k(\boldsymbol{x}_q, \boldsymbol{x}_p)$ plotted as a function of the Euclidean distance $\|\boldsymbol{x}_q - \boldsymbol{x}_p\|_2$ (using pairs of training points from the two Gaussian mixtures dataset). As outlined in Sec. 4 and SM B, the interpretation of an RBF kernel as Gaussian kernel that can be used in a KDE of $p(\boldsymbol{x})$ is important to justify implied OOD, at least for low-dimensional problems. Unfortunately, the kernels induced by common architectures do not seem to be distance-aware and are thus not useful for KDE.

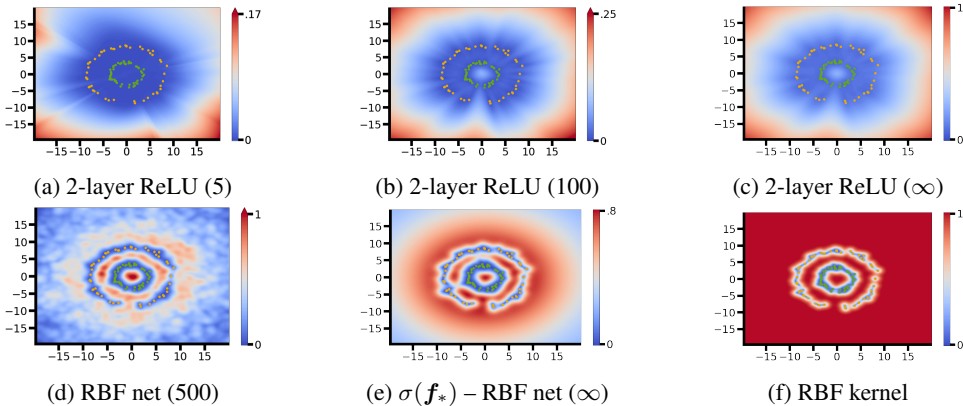

(a) 2-layer ReLU (5)      (b) 2-layer ReLU (100)      (c) 2-layer ReLU ($\infty$)

(d) RBF net (500)      (e) $\sigma(\boldsymbol{f}_*)$ – RBF net ($\infty$)      (f) RBF kernel

Figure S5: **Standard deviation $\sigma(\boldsymbol{f}_*)$ of the predictive posterior.** We perform Bayesian inference on a dataset composed by two concentric rings (SM D) comparing posterior uncertainties of GPs with NNGP kernels and an RBF kernel with those obtained by finite-width neural networks. Only the function space prior induced by an RBF kernel or RBF network causes epistemic uncertainties that allow outlier detection in the center or in between the two circles. However, the RBF network's uncertainties decrease with increasing input norm, which can be counteracted by further increasing $\sigma_\mu^2$ (Sec. 3).

## C.1   2D CLASSIFICATION

In this section, we consider the same dataset as in Fig 2 in a classification setting instead of regression. In this setting, exact inference is intractable even when using GPs and approximations are needed. In particular, we study classification with the logistic likelihood function $p(\boldsymbol{y} \mid f(\boldsymbol{x}; \boldsymbol{w})) = s(-\boldsymbol{y}f(\boldsymbol{x}; \boldsymbol{w}))$ where $s$ is the sigmoid function. We use HMC to sample from the posterior distribution in the finite-width limit, using a standard normal prior where the variance is inversely scaled by the hidden-layer's width. The results are reported in Fig S6 for ReLU, cosine and RBF networks.

As clearly depicted in the plots, our findings regarding the importance of the prior in function space can be extended also for the classification case. Indeed, the uncertainty captured by the ReLU architecture appears unsuitable for OOD detection given that the data distribution is not captured and the disagreement is low also in regions of the 2D plane that do not contain any training data points (also see Kristiadi et al., 2020; 2021). This pathology appears in both cases: when the disagreement is considered over the functions $\sigma(\boldsymbol{f}_*)$ or over the sigmoid outputs $\sigma(s(\boldsymbol{f}_*))$. By contrast, also for this new choice of likelihood, the cosine and RBF architecture exhibit uncertainty that conveys similar useful properties for OOD detection as seen in the regression example in Fig 2. Despite this empirical correspondence, it is important to note that the direct correspondence with the KDE technique that justifies the usefulness of OOD properties of the RBF kernel in the case of GP regression (see Sec. B) does not directly apply for classification. Indeed in the latter, we do not have an analytical form of the posterior as in Eq. 2 due to the non-Gaussianity of the logistic likelihood. Therefore our observations can not be readily generalized outside the settings of our experiments and further theoretical analysis are needed to asses an analogous justification for non-Gaussian likelihoods.

## C.2   SPLITMNIST REGRESSION

In this section, we consider SplitMNIST tasks (Zenke et al., 2017), which are binary decision tasks where the original MNIST dataset is split into five tasks containing two digits each: 0/1, 2/3, 4/5, 6/7, 8/9. To perform exact inference via Gaussian Processes, we consider each binary decision task as regression problem with labels -1/1. For computational reasons, the training set of each task is reduced to 1000 samples, but test sets remain at their original size. Results are reported in Table S1.

For the chosen length-scale parameters $l$, RBF kernels show best generalization. Though, they don't excel at OOD detection compared to other kernel choices. In SM B, we discussed that epistemic uncertainties as induced by the RBF kernel can be viewed as an approximation to $p(\boldsymbol{x})$ when the

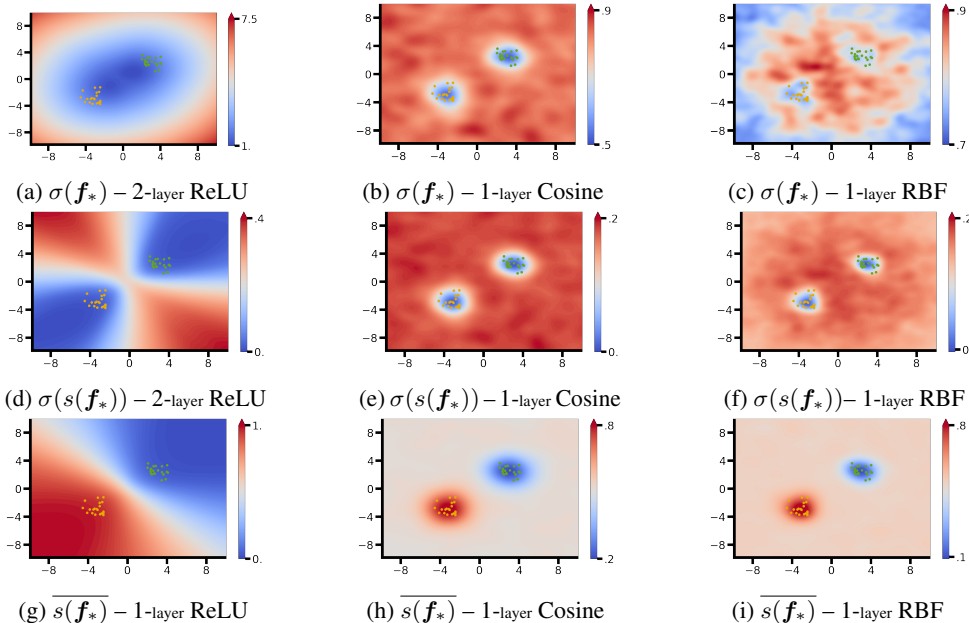

Figure S6: **Standard deviation over the logits $f_*$ and mean and standard deviation over the sigmoid outputs $s(f_*)$ for a 2D classification problem.** We perform approximate Bayesian inference with HMC for finite-width networks on a mixture of two Gaussians dataset considering different priors in function space induced by different architectural choices. The problem is now treated as classification task using HMC to sample from the posterior. We show the standard deviation $\sigma(f_*)$ of logits $f_*$ in (**a, b, c**), the standard deviation $\sigma(s(f_*))$ of the predicted probability for the input being in the positive class in (**d, e, f**), and the corresponding mean $\overline{s(f_*)}$ (**g, h, i**). The ReLU network has 5 hidden units, the cosine has 100 hidden units, and the RBF network has 500 hidden units.

Table S1: **GP regression on SplitMNIST tasks.** We perform GP regression on a single SplitMNIST task using the kernels reported in the first column (entries show mean and standard deviation when using different SplitMNIST tasks for training). AUROC values are computed when considering the test data from all remaining SplitMNIST tasks as OOD data, or by taking the test data of FashionMNIST as OOD.

|  | Acc. Train | Acc. Test | AUROC SplitMNIST | AUROC FashionMNIST |
|---|---|---|---|---|
| **RBF** ($l = 1$) | $100.0^{\pm .000}$ | $99.35^{\pm .577}$ | $.849^{\pm .077}$ | $.893^{\pm .062}$ |
| **RBF** ($l = 5$) | $100.0^{\pm .000}$ | $\mathbf{99.45^{\pm .552}}$ | $\mathbf{.892^{\pm .057}}$ | $.979^{\pm .010}$ |
| **RBF** ($l = 10$) | $100.0^{\pm .000}$ | $99.42^{\pm .505}$ | $.884^{\pm .060}$ | $.990^{\pm .005}$ |
| **ReLU** 1-layer | $99.74^{\pm .313}$ | $98.86^{\pm 1.01}$ | $.884^{\pm .060}$ | $.995^{\pm .002}$ |
| **ERF** 1-layer | $99.68^{\pm .356}$ | $98.76^{\pm 1.11}$ | $.884^{\pm .060}$ | $\mathbf{.996^{\pm .002}}$ |
| **Cosine** 1-layer | $99.68^{\pm .356}$ | $98.88^{\pm .985}$ | $.880^{\pm .062}$ | $.993^{\pm .003}$ |

dataset size goes to infinity $\mathcal{D} \to \infty$, and the length-scale goes to zero $l \to 0$. As expected, such approximation to $p(\boldsymbol{x})$ may deteriorate on high-dimensional image data, presumably as the Euclidean distance does not capture the geometry of the image manifold.

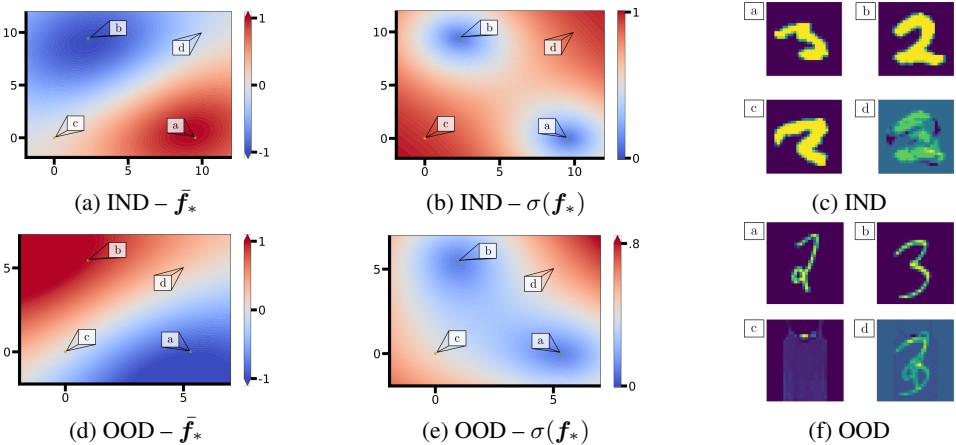

(a) IND $- \bar{\boldsymbol{f}}_*$     (b) IND $- \sigma(\boldsymbol{f}_*)$     (c) IND

(d) OOD $- \bar{\boldsymbol{f}}_*$     (e) OOD $- \sigma(\boldsymbol{f}_*)$     (f) OOD

Figure S7: **2D visualizations of the SplitMNIST posterior for a GP with RBF kernel** ($l = 5$). The plotted 2D linear subspaces are determined by 3 images (colored dots, denoted $a$, $b$ and $c$). In the upper row, the yellow dot represents the in-distribution (IND) test sample with the highest uncertainty (see image $c$ in subplot **(c)**), while the two green dots are the (in-distribution) training samples with smallest Euclidean distance to image $c$. In the lower row, the yellow dot represents the OOD test image with lowest uncertainty (see image $c$ in subplot **(f)**). Again, the two green dots are the closest IND training samples. Image $d$ is in both cases a randomly chosen point on the 2D subspace. Subplots **(a)** and **(d)** show the posterior mean $\sigma(\boldsymbol{f}_*)$, and subplots **(b)** and **(e)** the posterior's standard deviation $\sigma(\boldsymbol{f}_*)$. Subplots **(c)** and **(f)** show the images corresponding to the 4 highlighted points on the 2D subspaces.

To gain a better intuition on why the RBF kernel does not excel in OOD detection for image data, we visualize the uncertainty behavior in Fig. S7. The figure shows 2D linear subspaces of the 784D image space. These subspaces are determined by three images (see caption for details). As can be seen in Fig. S7b and S7e, a too small length-scale might cause test points to be not included in the low-uncertainty regions. On the other hand, if the length-scale is set too high, OOD points may fall inside low-uncertainty regions. The situations depicted in the figure show that for the given training set there is no trade-off length-scale $l$ that prevents such behavior (because some OOD points have less Euclidean distance to training points than some test points). Using an RBF kernel with a metric that encodes similarities in image space rather than using Euclidean distance might overcome these problems.

### C.3 CONTINUAL LEARNING VIA UNCERTAINTY-BASED REPLAY

In Sec. 7 we mention that the desideratum of having low uncertainty only on in-distribution inputs opens the possibility of considering uncertainty as an energy function from which we can sample. Thus, the uncertainty landscape can be used to construct a generative model.

In this section, we use continual learning (Parisi et al., 2019) to demonstrate this conceptual idea. In continual learning, a sequence of tasks $\mathcal{D}^{(1)}, \ldots, \mathcal{D}^{(T)}$ is learned sequentially such that each task has to be learned without access to data from past or future tasks. A common approach to continual learning is via replay (Lin, 1992). In this case, the current task $t$ is learned with the available training data $\mathcal{D}^{(t)}$ as well as datasets $\tilde{\mathcal{D}}^{(1)}, \ldots, \tilde{\mathcal{D}}^{(t-1)}$ which are supposed to represent the data distributions of the previous tasks. In the simplest case, $\tilde{\mathcal{D}}^{(s)}$ is constructed by storing a subset of $\mathcal{D}^{(s)}$ (Lin, 1992). Other approaches learn a separate generative model that can be used replay (fake) data $\tilde{\mathcal{D}}^{(s)}$ while a new task is learned (e.g., Shin et al., 2017; von Oswald et al., 2020)

We consider this (pseudo-)replay scenario without the need of maintaining a separate generative model. In Fig. S8, we consider a regression problem split into two tasks. Note, that learning a series

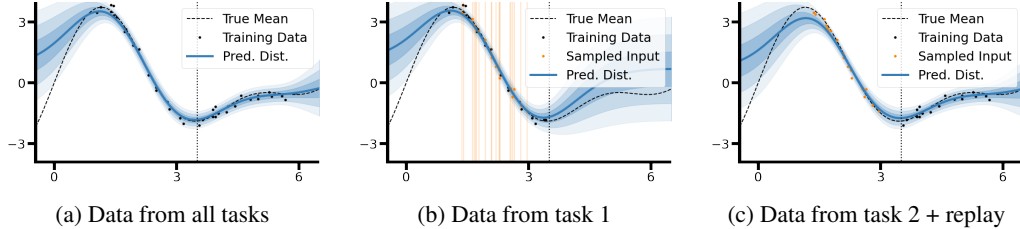

(a) Data from all tasks       (b) Data from task 1       (c) Data from task 2 + replay

Figure S8: **Continual learning with uncertainty-based replay.** As mentioned in Sec. 7, if uncertainty should only be low on in-distribution data, a generative model can be obtained by sampling from regions of low uncertainty. Here, we use this idea to realize a replay-based continual learning algorithm (Lin, 1992) without the need of storing data or maintaining a separate generative model. In this case, we split a polynomial regression dataset into two tasks (denoted by the black vertical bar). The posteriors in this figure are obtained via GP regression using an NNGP kernel corresponding to a 1-layer Cosine network. **(a)** All data is seen at once (no continual learning). **(b)** Only data of task 1 is seen to obtain the posterior. The orange vertical lines denote input locations sampled from low uncertainty regions via rejection sampling. Task 1's posterior is then used to label those input locations (orange dots). **(c)** The replay samples generated with the model of task 1 (orange dots) are used together with the training data of task 2 to obtain a combined posterior over all tasks.

of 1D regression problems is challenging for most continual learning algorithms, especially those relying on the recursive Bayesian update: $p(\boldsymbol{w} \mid \mathcal{D}^{(1)}, \mathcal{D}^{(2)}) \propto p(\mathcal{D}^{(2)} \mid \boldsymbol{w}) p(\boldsymbol{w} \mid \mathcal{D}^{(1)})$ (Henning et al., 2021). By contrast, (pseudo-)replay methods can solve this task with ease as the structure of the data in such low-dimensional problem is easy to capture by a generative model.

In our experiment, two tasks are created by splitting the dataset of a 1D function into two parts as illustrated in Fig. S8a. The left half is considered the first task, and the right half the second task, respectively. The blue posteriors shown in all three subpanels are obtained with a GP using the NNGP kernel of a 1-layer Cosine network, which should exhibit OOD properties similar to an RBF kernel (cf. Sec. 4). In Fig. S8a, the posterior is obtained using the combined datasets from both tasks (no continual learning). In Fig. S8b, the posterior is obtained using only the training data of the first task. After this (first task's) posterior is obtained, we can use it to perform pseudo-replay as outlined below to generate $\tilde{\mathcal{D}}^{(1)}$ which can be mixed with the data of the second task $\mathcal{D}^{(2)}$ to produce the posterior in Fig. S8c. The posterior in Fig. S8c looks qualitatively similar to the one in Fig. S8a even though it has been obtained without direct access to the first task's training data $\mathcal{D}^{(1)}$.

To generate $\tilde{\mathcal{D}}^{(1)}$ while learning the second task, we need access to the posterior of the first task. Note, the model from the previous task is often kept in memory by continual learning algorithms, for instance, to use it for regularization purposes (Kirkpatrick et al., 2017) or to replay data with previously trained generative models (Shin et al., 2017).[7] We generate in-distribution input locations (denoted by orange vertical bars in Fig. S8b) via rejection sampling as in Fig. 7. For that, we define epistemic uncertainty as an energy function $E(\boldsymbol{x}_*) \equiv \sigma(\boldsymbol{f}_*(\boldsymbol{x}_*))$ and construct a Boltzmann distribution $\tilde{p}(\boldsymbol{x}_*) \propto \exp(-E(\boldsymbol{x}_*)/T)$, where we set the temperature $T = 1$.[8] As uncertainty of the posterior of Fig. S8b is only low for in-distribution inputs, we can assume obtained input locations are similar to those represented in $\mathcal{D}^{(1)}$. To construct a dataset $\tilde{\mathcal{D}}^{(1)}$, we use the posterior of Fig. S8b to sample predictions $\boldsymbol{y}_*$ (orange dots in Fig. S8b and Fig. S8c).

---

[7]As we use a non-parametric model (a GP) which is represented by the training data $\mathcal{D}^{(1)}$, keeping the model in memory requires to store $\mathcal{D}^{(1)}$ too, which is a violation of continual learning desiderata. However, this is just a conceptual example. If the same experiment would be performed via approximate inference in a parametric model (e.g., HMC on a corresponding finite-width network), then the (approximate) posterior of the first task can be stored without storing $\mathcal{D}^{(1)}$. In the case exact inference can be performed, the Bayesian recursive update yields a sufficient continual learning algorithm.

[8]Note, if $E(\boldsymbol{x}_*) \propto -\log p(\boldsymbol{x}_*)$, then this process yields exact samples from the input distribution. For instance, considering the limiting case of SM B where $\sigma(\boldsymbol{f}_*(\boldsymbol{x}_*))^2 \approx 1 - p(\boldsymbol{x}_*)$, a reasonable choice of energy could be $E(\boldsymbol{x}_*) \equiv -\log [1 - \sigma(\boldsymbol{f}_*(\boldsymbol{x}_*))^2])$.

In summary, this section presents an example application that can arise if the desideratum of high uncertainty on all OOD inputs (i.e., robust OOD detection) is fulfilled by a Bayesian model. Whether BNNs can be constructed to fulfill this desideratum on real-world data is, however, unclear at this point.

## D   EXPERIMENTAL DETAILS

In this section, we report details on our implementation for the experiments conducted in this work. In all two-dimensional experiments: the likelihood is fixed to be Gaussian with variance $0.02$. Unless noted otherwise, the prior in weight space is always a width-aware centered Gaussian with $\sigma_w^2 = 1.0$ and $\sigma_b^2 = 1.0$ except for the experiments involving RBF networks where we select $\sigma_w^2 = 200.0$. The RBF kernel bandwidth was fixed to $1.0$ in all corresponding experiments.

**1D regression.**   We construct the 1D regression example (e.g., Fig 1a) by defining $p(\boldsymbol{x})$ uniformly within the ranges $[1.0, 1.3]$, $[3.5, 3.8]$ and $[5.2, 5.5]$, and $p(\boldsymbol{y} \mid \boldsymbol{x})$ as $f(x) + \epsilon$ with $f(x) = 2\sin(x) + \sin(\sqrt{2}x) + \sin(\sqrt{3}x)$ and $\epsilon \sim \mathcal{N}(0, 0.2^2)$. The training set has size 20.

**Periodic 1D regression (only Fig. 6).**   We construct this 1D regression example by defining $p(\boldsymbol{x})$ uniformly within the range $[0.0, 12.5]$, and $p(\boldsymbol{y} \mid \boldsymbol{x})$ as $f(x) + \epsilon$ with $f(x) = \sin(x)$ and $\epsilon \sim \mathcal{N}(0, 3^2)$. The training set has size 30.

**Gaussian Mixture.**   We created a two-dimensional mixture of two Gaussians with means $\mu_1 = (-2, -2)$, $\mu_2 = (2, 2)$ and covariance $\Sigma = 0.5 \cdot \mathbb{I}$ and sampled 20 training data points.

**Two rings.**   We uniformly sampled 50 training data points from two rings centred in $(0, 0)$ with inner and outer radii $R_{i,1} = 3, R_{o,1} = 4, R_{i,2} = 8, R_{o,2} = 9$, respectively.

**HMC.**   We use 5 parallel chains for 5000 steps with each constituting 50 leapfrog steps with stepsize $0.001$ for width 5 and $0.0001$ for width 100. We considered a burn-in phase of 1000 steps and collected a total of 1000 samples.

