# OpenReview forum: "Uncertainty-based out-of-distribution detection requires suitable function space priors"
_ICLR.cc/2022/Conference — ICLR 2022 Submitted_

### Official Review · Reviewer_fGuy · 2021-10-26

**Correctness:** 3
**Technical Novelty And Significance:** 2
**Empirical Novelty And Significance:** 1
**Recommendation:** 5
**Confidence:** 3

**Main Review:**

I believe this is an interesting paper. However, it lacks novelty in the sense that everything that is covered in the paper is more or less already known in the machine learning community. Furthermore, it is natural that the posterior predictive uncertainty is not suitable for OOD detection in some cases. The reason is that the posterior uncertainty can be high simply because there is no data in a particular region.

It is natural to expect that exact inference in infinite-width networks under common architectural choices does not necessarily lead to desirable OOD behavior. This is especially true if the model assumptions are wrong.

The idea that weight-space prior has a strong effect on OOD performance is something to be expected, since it will have a big impact in the posterior uncertainty.

I do not see why the use of incorporating prior knowledge, that is usually encoded in an input-domain agnostic manner, can negatively affect OOD uncertainties. It will reduce uncertainty, but that need not be bad for OOD detection. In fact, it can be beneficial.

Summing up, I believe that, although this paper shows some interesting concepts or ideas, it is does not provide a significant break thought in terms of analyzing OOD and the obtained results are more or less already known. This will limit its impact in the community.

I also the paper a bit misleading since in several figures the data presented for training is incompatible with the assumptions implied by the chosen prior. For example, in Figure 6 it seems that the RBF kernel is better for OOD detection since it will lead to higher uncertainties. I do not see this. In fact, it seems to me that it will be better to use the ESS kernel for that. (I mean OOD in the y values no the x values).

I also have the feeling that this paper is using the uncertainty on y to detect OOD in the x domain which is counterintuitive. I believe that if you want to detect OOD in the x domain you should have a model for x, not y given x, which is what BNN and GPs do.

The paper is more or less clear, however, and well written.


**Summary Of The Paper:**

This paper carries out an analysis that motivates that the use of the Bayesian predictive distribution and its uncertainty is not appropriate for detecting out of distribution data. The paper focuses on the case of Bayesian Neural networks and its infinite-wide generalization. Namely, a Gaussian process. The paper has no experiments but gives illustrative insights about why the Bayesian posterior is not suitable for detecting out of distribution data. They show that exact inference with GPs (infinitely wide networks) does not lead to desirable OOD detection. They discuss desirable kernel features for OOD. They emphasize that the choice of weight-space prior has a strong effect on OOD performance. They argue that there is a trade-off between good generalization and having high uncertainty on OOD.


**Summary Of The Review:**

A well written paper showing interesting concepts that are mostly well known in the machine learning community. The paper lack novelty.

---

> ### Author Response · Authors · 2021-11-17
> **Reply to reviewer fGuy**
>
> We thank reviewer fGuy for the provided feedback.
>
> Reviewer fGuy mostly criticizes the novelty of our work. We agree that our paper's content might in retrospect be obvious to any expert in Bayesian statistics. However, uncertainty-based OOD detection is omnipresent in today's deep learning literature, especially when approximate Bayesian methods are being presented. In this context, approximate Bayesian methods are often only marginally better (or even worse) than deterministic baselines in the task of OOD detection. We missed a discussion in the respective literature on the reasons for such poor performance. Since BNNs are often motivated in the context of safety-critical applications, it is important to understand (and transparently communicate) reasons for existing shortcomings of safety-critical desiderata such as OOD detection. Our hypothesis is that existing shortcomings are not being eliminated by improved approximate inference methods, but that the influence of modelling choices on OOD uncertainties need to be better understood to ensure sustainable progress in this field.
>
> For the above-mentioned reasons, we do not agree with the reviewer's statement that the discussed concepts are well known in the machine learning community.
>
> Regarding the question of whether to perform OOD detection on x or y. As stated in Sec. 2, we do not consider the problem of OOD detection within the training set (where we have access to tuples (x,y)), but we consider the OOD setting most often studied in the deep learning literature, where a deployed model (after training) is tested for the ability to recognize outlier inputs. As during deployment only inputs x are given to the system (and the system is asked to produce predictions y), we can only perform OOD detection on the observed inputs x.
>
> *"I also have the feeling that this paper is using the uncertainty on y to detect OOD in the x domain which is counterintuitive."*
> As the predictive distribution (a distribution over y) is induced by an input x, uncertainty derived from this distribution can be designed to be reflective of properties defined on the x-domain. However, the sole purpose of our paper is to raise awareness that such uncertainties are not automatically or intrinsically reflective of OOD properties.
>
> We hope our answer clarifies the positioning and impact of our paper, and convinces the reviewer to raise their score.

---

> > ### Comment · Reviewer_fGuy · 2021-11-24
> > **Response to Author's Rebuttal**
> >
> > I would like to thank the authors for the detailed response to my review. I appreciate the effort to improve the paper and to include extra experiments. Also, the clarifications they have provided. However, I still think that the paper's message does not represent a significant break thought in terms of analyzing OOD and the obtained results are more or less the expected ones. In any case, in appreciation for the author's effort, I have increased a bit my score.

---

> > > ### Author Response · Authors · 2021-11-28
> > > **Reply to reviewer fGuy's response**
> > >
> > > We thank reviewer fGuy for raising their score and appreciating our efforts towards incorporating the reviews.
> > >
> > > Reviewer fGuy still questions the scientific relevance of our work, and we would like to respond to this. Usually, when OOD performance is assessed in the literature, different methods are compared under identical experimental conditions (i.e., same network architecture and same weight prior; see, for instance, the reference list at the end of our introduction). We instead argue that it is (at least equally) important to vary the function space prior if OOD detection capabilities are assumed to arise due to uncertainties from the Bayesian posterior. Only by changing the prior, we can change our (subjective) definition of what we know and what we don't know. We therefore strongly believe that our paper carries a novel perspective and can guide future research in both, the design of methods for OOD detection and the way in which empirical evaluation is conducted. Now that this issue is exposed, we hope that more targeted research will lead to the discovery of function-space priors that empirically work well for OOD detection and thus provide a general incentive for intensified research on Bayesian deep learning.

---

> > > > ### Comment · Reviewer_fGuy · 2021-11-29
> > > > **Reply to the Authors**
> > > >
> > > > I would like to thank the authors for their response. I will keep my score as it is.

---

### Official Review · Reviewer_HkGA · 2021-11-01

**Correctness:** 3
**Technical Novelty And Significance:** 3
**Empirical Novelty And Significance:** 2
**Recommendation:** 6
**Confidence:** 4

**Main Review:**

The paper is well-written and clear, and I enjoyed reading it.
I think the topic is extremely timely and I fundamentally agree with the authors that there is not necessarily any good reason to think that BNNs should be good for OOD detection.
While the analysis does not really involve anything new from a technical level, the observations the authors make have important implications for the field and have not, to the best of my knowledge, really been discussed before.
However, there are aspects that I believe could be improved.
Therefore, I am willing to accept the paper, but only tentatively.

My main reservation is that the paper only truly considers regression tasks, as the classification task it does consider is done via regression.
While the authors do this to avoid having to use approximate inference, classification with classification likelihoods tends to perform quite differently to classification by regression.
This is particularly relevant, as most works assessing BNNs on OOD detection that I am aware of do so on classification, not regression.
It would be good if the authors could perform the same HMC using logistic regression on their toy problems.

Another reservation I have is that practically all the examples are low-dimensional and toy.
While I don't think this is strictly necessary for acceptance, since the paper makes no strong claims about more difficult problems, the paper would be significantly strengthened by using more complicated problems.

Finally, it would be good if the authors could comment more about OOD generalization.
This would be particularly relevant, as there has been a recent surge of interest in the performance of methods on OOD datasets such as CIFAR-10-C.
In particular, the authors propose a method for validating OOD properties in Sec. 7 as an alternative to these sorts of datasets.
However, the authors do not really provide many details as to how one could achieve this, nor do they provide any experiments where they actually use this proposal.
I think that an example of actually using this would be very useful in gauging the effectiveness of the authors' proposal.

There are also some references that it would be good if the authors could add or discuss, although this is more minor.
For instance, the NNGP equivalence for deep networks was derived concurrently with Lee et al. (2018) and more rigorously proven in [1].
The authors should also discuss the recent work of [2], which proposes a new BNN prior to improve OOD generalization.
Finally, the authors may find it interesting to discuss [3], which provides an explicit example of how the uncertainty of a BNN may fail in classification settings.


Minor points:
- Have the authors checked how well the HMC converges?
According to the experimental details, the HMC is only run for a relatively short amount of time, even accounting for parallel chains.


References:
[1] https://arxiv.org/abs/1804.11271
[2] https://arxiv.org/abs/2106.11905
[3] https://arxiv.org/abs/2010.02709

**Summary Of The Paper:**

In this work, the authors challenge the assumption that underlies many recent works that Bayesian neural networks should be well-suited to out-of-distribution detection.
In order to do this, the authors focus on a function-space view by examining the properties of infinite-width BNNs.
They use this analysis to argue that BNNs may not necessarily be well-suited to OOD detection.
They further argue that there is a tradeoff between OOD generalization and uncertainty, and finally propose an alternative method of validating OOD properties of models.

**Summary Of The Review:**

In summary, I found that this paper represents interesting and timely work that should inform future discussion as to how we think about OOD detection and generalization with BNNs.
However, I found the experimental evaluation somewhat lacking, and therefore am only tentatively recommending accept at the moment.
I look forward to reading the other reviews and to hearing the author responses.

---

> ### Author Response · Authors · 2021-11-17
> **Reply to reviewer HkGA**
>
> We thank the reviewer for the positive and encouraging assessment of our work. Below we address all points individually.
>
> We agree that experiments with **classification** tasks are a useful addition to the paper, and we added them in SM C.1. As pointed out by reviewer HkGA, we originally discarded them from the paper since our message concentrates on the fact that empirical OOD shortcomings can already be explained by considering the true posterior. Since we cannot study the true posterior in classification tasks, we cannot verify to what extend uncertainties are altered by the use of approximate inference. However, our regression plots (e.g., Fig. 2) suggest that approximate inference via HMC on such low-dimensional problems do not modify true uncertainties qualitatively. Hence, our new results suggest that also when using a classification likelihood, epistemic uncertainties are not necessarily reflective of the data-generating process.
>
> We applied the studied priors to a **high-dimensional** problem in SM C.2. As to be expected, the conclusions that can be drawn from these results are limited, but yet interesting as they add to the completeness of the paper. Note, the purpose of our paper is to question the intrinsic justification of BNNs for OOD detection in order to facilitate targeted research in this area. As we do not propose a prior that should lead to better OOD detection in practical settings, common OOD benchmarks with image datasets might not serve the message of our paper. Yet, we are open and grateful for any suggestion that could help us design experiments on high-dimensional data that serve the purpose of our paper (note, however, that we do not have the resources available to perform full-batch HMC on large high-dimensional datasets).
>
> **OOD generalization** is an important remark by reviewer HkGA. To us, this problem should first be addressed when deciding on a definition of OOD points (cf. Sec. 2). For instance, if Gaussian corruptions of sample points $x$ from the data manifold ($p(x)$) are permitted, then this should somehow be reflected in the definition of OOD. For a GP with RBF kernel, low uncertainty in a Euclidean ball around such sample points can be ensured by increasing the length scale. In general, generalization to such points can be facilitated in two ways. Either explicitly by enriching the dataset with such perturbations (data augmentation), or by incorporating this desideratum as prior knowledge (as done in [2]). In a sense, our Fig. 6 represents an extreme example of incorporating such prior knowledge, where periodicity over the whole $x$-domain is assumed such that observations within a small domain can be generalized to the whole real line. The problem of predicting under a covariate shift is, however, multifaceted, and we do not yet feel comfortable writing a detailed discussion about this topic in our paper. We now point the reader to the respective literature at the end of Sec. 6 and hope that future research can improve our understanding of how such desideratum can be reconciled with the Bayesian approach to learning.
>
> The proposal for **OOD validation** in Sec. 7 is more conceptual than practical. To illustrate the usefulness of this concept, we added a continual learning example in SM C.3. We additionally tried to sample from the predictive uncertainty via MCMC when considering a SplitMNIST task. However, such experiments failed in producing meaningful in-distribution images. Unfortunately, we cannot tell whether this is due to MCMC not converging to the stationary distribution or because the predictive uncertainty does not reflect the data manifold.
>
> **HMC convergence:** we repeated our experiments with HMC using 50000 steps per chain instead of 5000 and we couldn't notice any difference in the results. Moreover, during sampling with HMC we always monitored the energy of the chain in order to: (i) detect the burn-in phase and (ii) assess convergence to the stationary distribution.

---

> > ### Comment · Reviewer_HkGA · 2021-11-25
> > **Thank you for the response!**
> >
> > Thank you for your response! I am pleased to see the inclusion of results using classification likelihoods, as well as the SplitMNIST experiments. I think that these help to improve the quality of the paper. However, I would still argue that the experimental validation is a bit weak, and while I do sympathize with the authors’ remarks that they have limited computational resources, I think that this would need to be made up with by more theoretical results (as reviewer g5SX has suggested) or a more in-depth discussion of the relationship to OOD generalization/distribution shift, as this is one of the driving motivators of much recent work. Therefore, I remain hesitant to wholeheartedly recommend acceptance.
> >
> > Nevertheless, I would like to reiterate my comment that I think that this is an especially timely line of work and that this is an important discussion for the BNN community to have. My impression is that many works have (at least implicitly) assumed that BNNs will naturally be good at OOD-related tasks without providing reasons for this assumption. This paper tries to address this assumption explicitly. I think that given the fact that many works have used their methods on OOD tasks without addressing this assumption indicates that the discussion in this paper is at least relevant if not new in the literature.

---

> > > ### Author Response · Authors · 2021-11-28
> > > **Reply to reviewer HkGA's response (1/2)**
> > >
> > > We are encouraged by the fact that reviewer HkGA agrees on the relevance of the underlying message of our paper, and that this kind of discussion is necessary within the BNN community.
> > >
> > > We would like to use the opportunity and respond to the reviewer's remaining reservations with respect to our paper.
> > >
> > > **HMC experiments:** After additional considerations, we would like to further describe our view on the impact of additional high dimensional experiments. Apart from computational considerations, our conceptual concern is that these experiments bring no additional benefit to support our argument. Indeed as recently shown in [1], HMC on CIFAR leads to imperfect OOD detection, and reproducing such results would not allow us to disentangle whether these are due to the inability of the true posterior to be a perfect OOD detector, or because HMC is performing approximate inference and cannot reflect the exact posterior well.
> > > Furthermore, the fact that our SplitMNIST AUROCs are similar for different architectures (different function-space priors), further discourages a blind search for priors with HMC that lead to good OOD detection on real-world problems.
> > >
> > > **OOD generalization:** We do not want to argue against the relevance of this topic, and upon the reviewer's initial request we did add a paragraph to the end of Sec. 6, attempting to clarify our positioning. To us, OOD detection and OOD generalization are intrinsically related (generalization to OOD regions automatically suggests low epistemic uncertainty in these regions). As stated before, realizing OOD generalization has to be encoded in prior knowledge (if not explicitly ensured via data augmentation, which, however, would not justify the use of the word "OOD"). Thus, to obtain OOD generalization one would need to encode in the prior "where" and "how" generalization should occur. To link again to our Fig. 6b: In this example, the prior says that generalization should occur across the whole real line ("where") by repeating the periodic behavior extracted from the training data ("how"). This is a very simplistic example, but it hopefully illustrates the complexity associated with the problem of OOD generalization. By contrast, reliable OOD detection simply requires to have high uncertainty everywhere OOD. To summarize, under exact Bayesian inference both problems solely rely on the chosen function-space prior, but have somewhat opposing desiderata. We believe that OOD generalization deserves to be studied separately to avoid a further dilution of our paper's simple message.
> > >
> > > [1] Izmailov et al., "What Are Bayesian Neural Network Posteriors Really Like?", ICML, 2021.

---

> > > > ### Author Response · Authors · 2021-11-28
> > > > **Reply to reviewer HkGA's response (2/2)**
> > > >
> > > > **More theoretical analysis:**  The reviewer's response made us reflect on how to more convincingly convey our arguments using theoretical insights. We currently study the infinite-width limit to make conjectures about the finite-width limit due to intractability. However, there is the known special case of linear neural networks (without hidden layers) with Gaussian prior and likelihood where Bayesian inference is tractable. Also in this case, it becomes immediately obvious that the posterior variance does not reflect $p(x)$. We provide the general mathematical derivation and illustrations in the anonymous document linked below. For the sake of simplicity, let's look at the posterior in function space if $y$ is a scalar:
> > > >
> > > > $p(f_* \mid x_*, X, y) = \mathcal{N} \big( f_* ; (x_*)^T \mu_\text{post}, (x_*)^T \Sigma_\text{post} x_* \big)$
> > > >
> > > > where $\mu_\text{post}$ and $\Sigma_\text{post}$ are the mean and covariance matrix of the posterior parameter distribution. Looking at the variance formula $(x_*)^T \Sigma_\text{post} x_*$ at a test point $x_*$ indicates that the variance depends on the norm of $x_*$ but does not, in general, reflect the data manifold of $p(x)$. For instance, lowest variance is attained at the origin $x_* = 0$, irrespective of whether the origin is an in-distribution or OOD point. Furthermore, when rotating the input coordinate system to be axis-aligned with the ellipse defined by $\Sigma_\text{post}$, posterior variance can simply be written as $\sum_i \sigma_{\text{post},i}^2 (x_{*,i}^\text{rot})^2$. Thus, unless $p(x) = \mathcal{N}(0, \Sigma_\text{post})$, epistemic uncertainty (measured as variance over function values) is not useful for out-of-distribution detection.
> > > >
> > > > See the following document for more details:
> > > > [https://tinyurl.com/yc7yyb2z](https://www.dropbox.com/s/z48ywiqclqcxjwq/PredictivePosteriorOfLinearBNN.pdf?dl=0)
> > > >
> > > > Note, that such instantiation of linear BNNs essentially amounts to Bayesian linear regression (see Sec. 2.1.1 in [2]), and our outline above is consistent with the following comment by Rasmussen:  "The predictive variance is a quadratic form of the test input with the posterior covariance matrix, showing that the predictive uncertainties grow with the magnitude of the test input, as one would expect for a linear model".
> > > >
> > > > This remark on linear BNNs might be perceived as obvious in retrospect, but it further contributes to our assessment that there is no a priori reason to believe that nonlinear BNNs are intrinsically good OOD detectors.
> > > >
> > > > If deemed interesting by the reviewer, we are happy to incorporate such derivation in our paper to strengthen our arguments.
> > > >
> > > > Again, we would like to express our gratitude towards reviewer HkGA for his constructive and thoughtful review.
> > > >
> > > > [2] Rasmussen, "Gaussian Processes in Machine Learning", Springer, 2004.

---

> > > > > ### Comment · Reviewer_HkGA · 2021-11-29
> > > > > **Reply to the Authors**
> > > > >
> > > > > Thank you for the thoughtful reply. I understand the concerns the authors have about HMC, and I certainly would not be expecting them to run experiments similar to the compute-heavy HMC experiments performed in Izmailov et al. Indeed, even with the impressive lengths that Izmailov et al. went to it is impossible to say with certainty that their HMC converged.
> > > > >
> > > > > However, I still stand by my update that I share some of the concerns of the other reviewers that this work would need either a stronger empirical evaluation or some significant theory for me to wholeheartedly recommend acceptance more strongly - while this work poses an important question, to me it does not make significant enough progress in answering it. I respect the authors' decision to leave OOD generalization for future work, but in this case even more should be done along the lines of OOD detection.
> > > > >
> > > > > Finally, on a technical note, I would like to point out that it is not true that linear BNN posteriors are Gaussian and are in general intractable. An easy way to see this would be to consider that there are permutation symmetries in the posterior, meaning that the posterior is likely highly multimodal. Alternatively, it is clear that the function-space prior is not Gaussian as the product of Gaussian random variables is not Gaussian.
> > > > >
> > > > > I would like to thank the authors again for the interesting discussion, and I regret that I do not feel able to increase my score. I look forward to seeing how future iterations of this paper turn out.

---

> > > > > > ### Author Response · Authors · 2021-11-29
> > > > > > **Reply to reviewer HkGA**
> > > > > >
> > > > > > Dear reviewer HkGA,
> > > > > >
> > > > > > we thank you for your response and respect your decision.
> > > > > >
> > > > > > Just to reply to the technical note: Our above formulation should have been more careful, we only consider the case of having no hidden layers (see linked document). A reparametrization of a linear NN (e.g., by adding hidden layers) may destroy the posterior's Gaussianity, as pointed out by the reviewer. Our example was simply meant as a further illustration to highlight that there is no intrinsic justification that posterior uncertainties reflect the data-generating process.
> > > > > >
> > > > > > We will update the confusing sentence in our reply above!
> > > > > >
> > > > > > Thank you for the discussion!

---

### Official Review · Reviewer_g5SX · 2021-11-03

**Correctness:** 4
**Technical Novelty And Significance:** 2
**Empirical Novelty And Significance:** 2
**Recommendation:** 5
**Confidence:** 2

**Main Review:**

Strengths
1. The manuscript includes not only finite-width networks but also infinite-width networks in the analyses.
2. The manuscript points out potential problems when naively using BNNs for ODD detection.

Weaknesses
1. The manuscript does not provide any theoretical analyses for BNNs on OOD problems.
2. All the conclusions are made from toy simulated datasets.

**Summary Of The Paper:**

The manuscript challenges the widely believed assumption that Bayesian neural networks (BNNs) are well suit for out-of-distribution (OOD) detection, by showing empirical results obtained using infinite-width (allowing the exact inference, because a network can be equivalently represented by a GP with the included kernel) and finite-width networks (requiring the approximate inference). The manuscript provides several observations in line with this purpose (to disclose potential problems when BNNs are used for ODD detection).  For example, the exact inference in infinite-width networks does not necessarily lead to desirable OOD behavior, e.g., the posterior function standard deviation obtained from an infinite-width 2-layer ReLU does not reflect the underlying data generating process and this uncertainty is not suit for OOD detection. In addition, this observation is consistent with the corresponding finite-width networks.

**Summary Of The Review:**

The manuscript has a certain contribution that it points out potential problems when naively using BNNs for ODD detection. However, I think that this contribution is not enough due to the same reasons I listed as the weaknesses of this study above.

---

> ### Author Response · Authors · 2021-11-17
> **Reply to reviewer g5SX**
>
> We thank reviewer g5SX for their review. Reviewer g5SX stresses two aspects of our paper, which they consider weaknesses. We would like to answer to both points separately below.
>
> *1. "The manuscript does not provide any theoretical analyses for BNNs on OOD problems."*
>
> We agree that theoretical progress is needed to justify the use of BNNs for OOD detection. However, the message of our paper is to question the common assumption that BNNs are intrinsically good for OOD detection, and it is not obvious to us how an additional theoretical analysis could strengthen that message.
>
> Any such analysis will first require a clear definition of what is meant by an OOD point (cf. Sec. 2). For instance, we consider the simple desideratum that uncertainty should be inversely related to $p(x)$ and generally  high outside the support of $p(x)$ (and the manifold on which $p(x)$ is defined). When we observed that epistemic uncertainty fulfills this desideratum in our low-dimensional problem settings, we gave an explanation for how uncertainty can be related to $p(x)$ (e.g., see SM B). If epistemic uncertainty did not reflect $p(x)$ (using proper Bayesian inference), it is likely not possible to show such relation. Moreover, the connection we established between the variance of the posterior of a GP with RBF kernel and the KDE technique to explain the uncertainty, is, despite its simplicity, a mathematical result that holds independently on the complexity (dimensionality) of the experiments we performed. We agree that in more complicated experiments the benefits underlined by this link might not be evident nor explicit but we consider this being solely a problem of the distance measure considered in the space and used to formulate the RBF kernel (see SM C.2).
>
> We are open and grateful for any suggestion on how to further improve the clarity of our paper with theoretical arguments.
>
> *2. "All the conclusions are made from toy simulated datasets."*
>
> Our paper aims to provide a clear message. An abundance of OOD experiments with real-world data can be found in the literature. The results in these experiments show that BNNs somehow could work for OOD detection, but that the performance is often not satisfactory. The fundamental reasons for such poor performance are never discussed to the best of our knowledge, or are attributed to the use of approximate inference.
> In our work we instead point out how the performance shortcomings are due to a fundamental lack of evidence that connects OOD detection and Bayesian inference in the first place.
>
> We believe that progress in this field can be facilitated if this lack is overcome and the foundations of this connection are more thoroughly discussed and studied. We agree, however, with the reviewer that for the sake of completeness the reader benefits from seeing how the particular priors discussed in our work are performing on high-dimensional data. Therefore, we added results on MNIST in C.2.

---

### Author Response · Authors · 2021-11-17
**General response to all reviewers**

We thank all reviewers for the time taken to assess our manuscript and for the valuable feedback and suggestions.

We updated the paper and incorporated suggestions from the reviewers. We summarize major modifications below and address individual reviewer concerns in point-by-point responses.

We added three sections to the SM with additional experiments that were requested by the reviewers.

In SM C.1, we added experiments with a likelihood suitable for binary classification problems. While reported results are all obtained via (high-fidelity) approximate inference, they are consistent with the remaining experiments by showing that even under such likelihood, common architectural choices are not suitable for OOD detection.

In SM C.2, we consider a simple image dataset by constructing a binary decision problem on MNIST. This is an example of high-dimensional data that has no intrinsic Euclidean structure. The unknown structure of the data makes it difficult to determine desirable function space properties for OOD detection, and thus further emphasizes the need for studying the relation between the exact Bayesian posterior and the problem of OOD detection.

Furthermore, we added a continual learning experiment to SM C.3, demonstrating potential applications that are implied by the desideratum on having low predictive (epistemic) uncertainty only on in-distribution inputs. The purpose of this experiment is not to propose a new strategy for continual learning, but to emphasize that the aforementioned desideratum essentially requires the Bayesian posterior to perform generative modelling (rather than just solving the discriminative task at hand).

We hope these additions and our responses convince the reviewers about the relevance of our work.

We remain available to answer any further questions or elaborate more extensively if our clarifications are not yet sufficient.

---

### Decision · Program_Chairs · 2022-01-20

**Decision:**

Reject

**Comment:**

The authors question the assumption that the epistemic uncertainty provided by Bayesian neural networks should be useful for out of distribution detection. They start their analysis in the infinite width limit so as to be able to understand how the induced kernels in a Gaussian process behave. The paper also discusses the potential tradeoffs between generalization and detection. Overall, the paper presents some facts that, while not surprising, (Reviewer fGuy), are helpful in questioning the default assumption. Overall, though, the combination of the lack of surprise with the multi-part, somewhat loosely connected message reduces the quality of the submission.